# Secretin targets interstitial cells of Cajal to regulate intestinal contractions

Allison M Bartlett ⬛, Peter J Blair, Kenton M Sanders ⬛ ✉ & Salah A Baker ⬛ ✉

## Abstract

Secretin is a gastrointestinal (GI) hormone that slows intestinal motility, an effect thought to be mediated through vagal afferent pathways. In this study we show evidence for a novel function of secretin involving a non-neural mechanism mediated by interstitial cells of Cajal (ICC). Transcripts of secretin receptors (*Sctr*) are expressed abundantly by ICC in the deep muscular plexus (ICC-DMP). Secretin inhibits small intestinal contractions in the presence of the neurotoxin, tetrodotoxin (TTX) and suppresses excitatory enteric neurotransmission. The inhibitory effects of secretin occur through inhibition of $Ca^{2+}$ transients in ICC-DMP, likely via Gαs-coupled cAMP production and PKA activation that leads to inhibition of IP3 receptors. Our results provide a novel concept for the role of ICC-DMP in small intestinal motility. ICC-DMP serve as integration hubs in which signaling from the enteric nervous system and hormones converge and integrate regulatory responses controlling intestinal motility. In the case of secretin, integrated responses may serve to slow intestinal transit to enhance digestion and absorption of nutrients.

**Keywords** Secretin; Secretin Receptor; Ca2+ Imaging; Intestinal Motility; cAMP

**Subject Category** Digestive System

## Introduction

Secretin, a member of the secretin-glucagon-vasoactive intestinal peptide hormone superfamily, is a multifunctional gastrointestinal (GI) peptide hormone. It is primarily secreted postprandially during the intestinal phase of digestion by duodenal enteroendocrine S cells located in the crypts of Lieberkühn (Roth and Gordon, 1990). Once released into the circulation, secretin acts on a range of target tissues in both the central nervous system and the periphery (Murphy and Bloom, 2006). Secretin is known for its role in regulating bicarbonate secretion from pancreatic ducts to neutralize acidic chyme entering the duodenum (Hootman and de Ondarza, 1993; Ishiguro et al, 2012). However, its broader physiological effects include the regulation of metabolism and energy expenditure (Schnabl et al, 2021; Sekar and Chow, 2013), osmoregularity

(Bai and Chow, 2017; Chu et al, 2007), thermogenesis in adipose tissue (Li et al, 2018), prevention of obesity (Laurila et al, 2021) and in GI motility (Lu and Owyang, 2009; Li et al, 1998) (Gutiérrez et al, 1974). Several studies have elucidated the mechanisms by which secretin regulates gastric motility and secretion but its effects on intestinal motility remain incompletely understood. Current research suggests that secretin's inhibitory effects on motility may be due to effects mediated by vagal afferent pathways and inhibition of motilin release (Kwon et al, 1999; Li et al, 1998; Lu and Owyang, 1995; Raybould and Holzer, 1993). Additionally, a single-cell transcriptomic study found expression of secretin receptors in human enteric neurons (Drokhlyansky et al, 2020), although it has not been determined whether effects of secretin on motility are mediated by enteric neurons or by other cell types.

The secretin receptor (SCTR) is a G protein-coupled receptor (GPCR) expressed in various target organs including the pancreas, liver, stomach, intestines, heart, and regions of the brain (Zaw et al, 2019) (Chu et al, 2006; Wang and Zhang, 2020). Upon secretin binding, SCTRs couple to effects through Gαs to stimulate adenylyl cyclase (AC), increase intracellular levels of cyclic adenosine monophosphate (cAMP) and activate downstream effectors, such as protein kinase A (PKA) or the exchange protein activated by cAMP (EPAC) (Ulrich et al, 1998) (Chepurny et al, 2012; Gespach et al, 1986; Ramos-Alvarez et al, 2019). SCTR activity is tightly regulated by desensitization through phosphorylation of its C-terminal tail (Shetzline et al, 1998) and subsequent receptor internalization (Holtmann et al, 1996).

In addition to neural regulation, a complex network of cells provides additional regulation of motor behaviors, such as setting of basal excitability and generation of electrical pacemaker activity (Sanders et al, 2023; Sanders et al, 2014). This network, postjunctional from the varicose nerve terminals of enteric motor neurons, is composed of smooth muscle cells (SMCs), interstitial cells of Cajal (ICC) and platelet-derived growth factor receptor alpha-positive (PDGFRα+) cells. These cells are electrically coupled, and the network has been referred to as the "SIP" syncytium (Sanders et al, 2023; Sanders et al, 2014). Two populations of ICC have been described in the small intestine: one population lies within the plane of the myenteric plexus (ICC-MY), and another lies near the submucosal surface of the circular muscle (CM) within the deep muscular plexus (ICC-DMP). ICC-MY are pacemaker cells that generate and actively propagate electrical slow waves, and ICC-DMP are neuromodulators that transduce neural inputs and are closely associated with enteric motor neurons. Because of the electrical coupling with SMCs,

Department of Physiology and Cell Biology, University of Nevada, Reno School of Medicine, Reno, NV 89557, USA. ✉E-mail: ksanders@med.unr.edu; sabubaker@med.unr.edu

transmembrane ionic currents activated within ICC can conduct to SMCs and thus regulate excitability. Expression studies have shown that small intestinal ICC express a variety of receptors for neurotransmitters and other regulatory substances, such as secretin (Lee et al, 2017). However, very low expression of *Sctr* was observed in SMCs and PDGFRα+ cells. Earlier gene array studies showed that *Sctr* was expressed dominantly by ICC-DMP (Chen et al, 2007). These findings suggested the hypothesis that ICC-DMP might be a novel target for secretin in regulation of small intestinal motility. In this study, we have investigated the role of secretin in regulating contractions of mouse and monkey jejunum. Our results suggest that secretin regulates Ca²⁺ transients in ICC-DMP by a novel mechanism.

## Results

### Secretin reduces the force of contraction in small intestinal tissues

We tested the effects of secretin on the contractile behavior of murine jejunal muscles before and in the presence of TTX (Fig. 1A,B). Under control conditions, secretin (100 nM) reduced spontaneous contractions area under the curve (AUC) to 45.6% ± 3.71 mN*min of control (Fig. 1A,C; Appendix Table S1) and decreased the contractile amplitude to 61.3% ± 4.42 (mN) (Fig. 1D). Secretin had no effect on the frequency of contractions ($P = 0.06$; Fig. 1E). The inhibitory effects of secretin (100 nM) were similar after the addition of the neurotoxin, TTX (1 μM). In the presence of the TTX, secretin reduced AUC to 46.4% ± 4.51 mN*min (Fig. 1B,C) and amplitude of contractions to 53.3% ± 3.37 mN (Fig. 1D). Secretin did not elicit a significant change in frequency in the presence of TTX ($P = 0.34$; Fig. 1E) and there were not any significant differences between responses to secretin with or without the addition of TTX. Lower concentrations of secretin (5 nM and 10 nM) were also tested but did not yield optimal effects (Appendix Fig. S1). The inhibitory effects of secretin on jejunal muscle contraction were restored quickly after a washout (Appendix Fig. S2).

We also compared the effects of secretin on jejunal muscle strips from Macaca fascicularis, a non-human primate (Fig. EV1). Secretin (100 nM) inhibited muscle contractions AUC to 59.3% ± 8.92 (mN*min) (Fig. EV1A,B; Appendix Table S2), reduced the amplitude of contractions to 45.8% ± 9.0 (mN) (Fig. EV1C) and had no significant effect on contractile frequency ($P = 0.68$; Fig. EV1D). In macaca muscle strips, the effects of secretin on jejunal contractions were similar after the addition of TTX (Fig. EV1E–H). Secretin (100 nM) inhibited muscle contractions AUC to 59.6% ± 7.29 (mN*min), amplitude to 51.0% ± 9.05 (mN) and caused no change in contractile frequency ($P = 0.2412$; Fig. EV1H). These data demonstrate that secretin regulates intestinal muscle contractions of mice and non-human primates independent of effects mediated by neuronal inputs.

### Secretin receptor (*Sctr*) is expressed in small intestinal ICC-DMP

Secretin inhibited intestinal contractions before and after addition of TTX, but specific cellular targets for the post-junctional effects of

secretin are difficult to determine using intact strips of muscle. RNAseq data obtained from ICC purified by cell sorting showed that small intestinal ICC display relatively high expression of *Sctr* and minimal expression in SMCs and PDGFRα+ cells (Lee et al, 2017), suggesting that responses to secretin may be transduced through ICC. The fact that secretin inhibited contractions without affecting contractile frequency suggests that this hormone may have variable effects on the different populations of ICC. Effects of secretin on the amplitude of contractions might indicate effects on either ICC-MY or ICC-DMP, but the lack of effects on frequency suggest that these effects are not mediated by the pacemaker cells, ICC-MY. Therefore, we performed in situ hybridization using RNAscope for *Sctr* on cross sections to characterize the spatial distribution of secretin receptor expression in jejunal muscles (Fig. 2). In these experiments Brightfield was utilized to visualize the circular muscle layer, DMP layer and the myenteric layer (Fig. 2A). DAPI labeling was used to visualize nuclei (Fig. 2B), and immunostaining with c-Kit antibody was used to identify ICC (Fig. 2C). RNAscope (Fig. 2D) showed that *Sctr* is expressed predominantly in ICC-DMP (Fig. 2E,F). *Sctr* was abundantly expressed in ICC-DMP but not resolved in ICC-MY as visualized in flat-mounts of the deep muscular plexus (Fig. 2G) and plane of the myenteric plexus (Fig. 2H). In ICC-DMP 95% of cells expressed *Sctr* with an average of 19.79 ± 1.53 puncta per cell (Fig. 2I, $N = 17$). Based on ACD Bio grading guidelines, a score of "3" was assigned to and score of "0" was assigned to ICC-MY (see "Methods" for details of this grading) (Fig. 2I). Negative controls did not show significant representation of RNAscope punctate, or clusters associated with individual cells although there was minor background noise (Appendix Fig. S3). These results indicate that the primary site of secretin receptor expression is ICC-DMP in the circular muscular layer of the murine small intestine.

### Secretin inhibits Ca²⁺ transients in ICC-DMP cells

ICC-DMP exhibit spontaneous Ca²⁺ transients that are stimulated by excitatory neural inputs and inhibited by inputs from enteric inhibitory neurons (Baker et al, 2018a; Baker et al, 2016; Baker et al, 2018b). Ca²⁺ release in ICC couples to activation of inward current through the Ca²⁺-activated Cl⁻ conductance channel ANO1. Inhibition of Ca²⁺ transients and ANO1 current therefore provides a net inhibitory influence on contraction. Therefore, we tested the effects of secretin on Ca²⁺ signaling in ICC-DMP, where *Sctr* was expressed, and in ICC-MY that showed scant expression of *Sctr*. For these experiments, we used confocal microscopy and tissues of mice with GCaMP6f expressed exclusively in ICC (Fig. 3A). ICC-DMP fired localized, stochastic, Ca²⁺ transients by Spatio-temporal Ca²⁺ maps (STMaps) (Fig. 3B). Secretin (100 nM) inhibited Ca²⁺ transients in ICC-DMP in the presence of TTX (1 μM) to block possible neuronal influences (Fig. 3A–C; Expanded view Movie EV1). In most experiments Ca²⁺ transients were completely inhibited upon perfusion with secretin for 3 min, however in a few experiments a few Ca²⁺ transients persisted in the presence of secretin but were significantly reduced to 3.1 ± 1.1 (per 30 s) in comparison to periods prior to secretin addition 10.5 ± 1.2 (per 30 s) (Fig. 3D). Other characteristics of Ca²⁺ transients were also significantly reduced: areas of Ca²⁺ transients were reduced to 0.57 ± 0.2 (μm*s) (Fig. 3E), duration of Ca²⁺ decreased to 1.20 ± 0.39 (ms) (Fig. 3F) and spatial spread decreased to

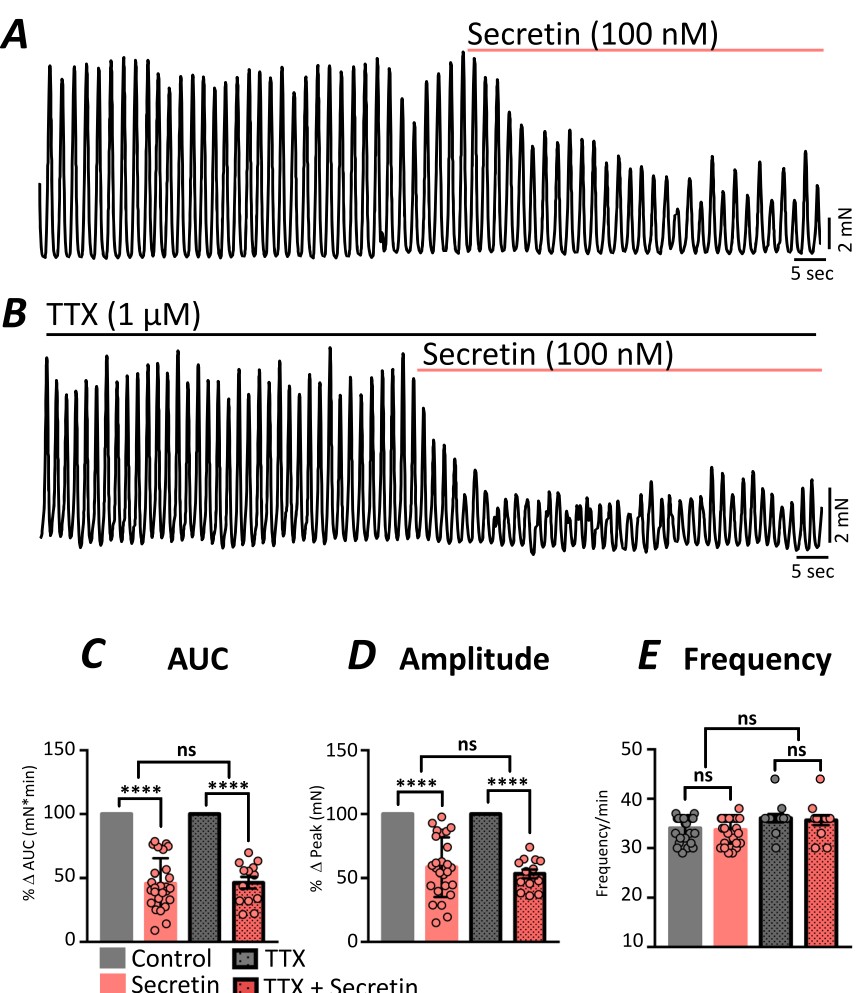

**Figure 1. Secretin inhibits murine small intestinal muscle contractions.**

(A) Contractions of jejunal muscle strips were reduced by application of secretin (100 nM). (B) The effects of secretin were unchanged in the presence of TTX (1μM). Contraction parameters were indicated as area under the curve (AUC), amplitude and frequency. (C) Area under the curve (AUC; mN*min) and (D) amplitude (mN) were reduced by secretin, but (E) the frequency (min⁻¹) of contractions was unaffected under control conditions and in the presence of TTX. All data were normalized to control responses, except the contraction frequency. The data are plotted as mean ± SEM, and significance was determined using paired $t$ test, ****$P < 0.0001$ and ns = 0.9167 for (C); ****$P < 0.0001$ and ns = 0.4599 for (D); and ns = 0.0589, 0.3370 and ns = 0.0627, respectively for (E)). Strips = 35, $n = 8$ for control and strips = 13, $n = 7$ for TTX preparations. Source data are available online for this figure.

3.1 ± 1.1 (μm) (Fig. 3G; Appendix Table S3). Secretin also inhibited ICC-DMP Ca²⁺ transients under control condition (no drugs added, Fig. EV2). The frequency of Ca²⁺ transient events, area, duration, and spatial spread all significantly decreased with the addition of secretin in ICC-DMP in the absence of TTX (Fig. EV2A–F).

ICC-MY generated rhythmic Ca²⁺ transients that spread cell-to-cell across fields of view (Fig. 3H–N). Secretin (100 nM) did not affect Ca²⁺ transients in ICC-MY in the presence of TTX (Figs. 3I and 4J) and had little effect on any of the Ca²⁺ transient parameters analyzed: frequency ($P = 0.21$), Ca²⁺ event area ($P = 0.15$), duration ($P = 0.34$) or spatial spread ($P = 0.91$) (Fig. 3K–N; Appendix Table S3). These data are consistent with the spatial localization of *Sctr* transcripts and demonstrate a correlation between gene expression and presence of functional secretin receptors in ICC-DMP in small intestinal tissues. This data also provides a

mechanism for the inhibitory effects of secretin in muscle strip contraction experiments.

## Secretin dampens responses to excitatory neural inputs

The inhibitory effects of secretin are significant however, elevated levels of secretin do not occur in isolation and would overlap with input from enteric motor neurons. Therefore, we tested whether the inhibitory effects of secretin could be overwhelmed by excitatory neural inputs by examining the effects of electrical field stimulation (EFS) of intrinsic neurons before and in the presence of secretin. It was shown previously that enteric neural responses are mediated via ICC-DMP in the small intestine (Baker et al, 2018a; Iino et al, 2004). We tested whether the effects of secretin on Ca²⁺ signaling in ICC-DMP were affected by a background of continuous EFS (3 Hz). Secretin (100 nM) persisted in inhibiting Ca²⁺ transients in ICC-

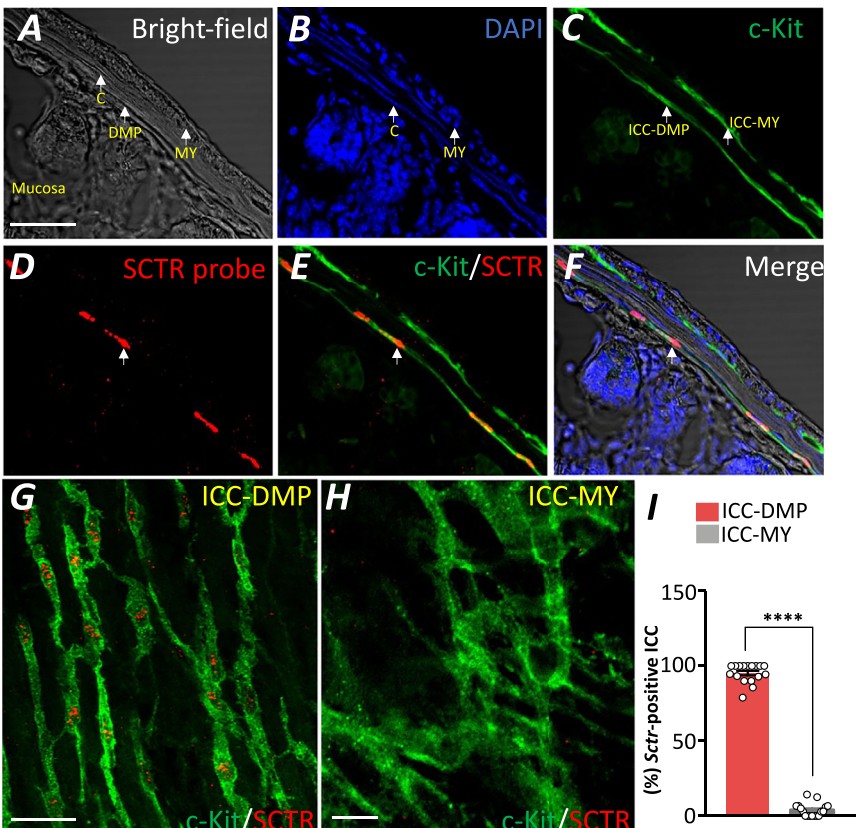

**Figure 2. Secretin receptor (*Sctr*) expression is abundant in ICC-DMP.**

(A) Brightfield confocal image of a cross-section of murine jejunum (×20). Arrows indicate circular muscle (C), deep muscular plexus (DMP) and myenteric plexus (MY) layers. ICC-DMP reside in the deep muscular plexus near the mucosal border of the circular muscle. (B) DAPI nuclear staining of section in (A). (C) Antibodies immunostaining of c-Kit (green) of ICC. White arrows indicate the location of ICC-DMP and ICC-MY. (D) visualization of the RNAscope probe for *Sctr* (red). (E) *Sctr* by RNAscope is localized in ICC-DMP cells as shown in the merged image of c-Kit and *Sctr* probe. (F) Merged images of B-D. (G) Jejunal flat mounts show circular muscle at the level of the deep muscular plexus region, where ICC-DMP (c-Kit) and *Sctr* probe are colocalized. (H) The *Sctr* probe was not localized to ICC-MY. (I) Summary graph showing the percentage of positive ICC cells for the *Sctr* probe (cKit⁺ cells and *Sctr* RNAscope punctate) in ICC-DMP vs. ICC-MY in jejunal cross-sections. ICC-DMP ACD score of 3, determined by punctate number per cell and cluster quantification. Scale bar represents 50 μm for images (A–F) and 25 μm for (G, H). The data are plotted as mean ± SEM, and significance was determined using paired *t* test, ****$P < 0.0001$ (I); cells = 244, *n* = 17 for ICC-DMP and for ICC-MY, cells = 397, *n* = 17. Source data are available online for this figure.

DMP under these conditions (Fig. 4A,B). Secretin inhibited Ca²⁺ transient frequency to 1.4 ± 0.46 from 26.16 ± 2.8 (Ca²⁺ transients/30 s) (Fig. 4C) and reduced other ICC-DMP Ca²⁺ transients' parameters: area of Ca²⁺ events 0.69 ± 0.2 (μm*s) (Fig. 4D), duration 0.99 ± 0.25 (ms) (Fig. 4E) and spatial spread 4.3 ± 1.2 (μm) (Fig. 4F) (Appendix Table S4). These results indicate that secretin inhibits Ca²⁺ signaling in ICC-DMP and superposition of neural inputs is incapable of rescuing Ca²⁺ release. However, during EFS stimulation inhibitory neurotransmitters are also released which could have amplified the negative effects of secretin on Ca²⁺ transients. Therefore, we also performed EFS (10 Hz for 10 s) before and after secretin in the presence of L-NNA (100 μM; NO synthase antagonist) and MRS2500 (1 μM; P2Y1 receptor antagonist), to inhibit major enteric inhibitory neurotransmission. We increased the frequency of EFS in these experiments to strongly activate excitatory neural drive on ICC-DMP, as previously documented (Baker et al, 2018a; Baker et al, 2018b). In control conditions, EFS in the presence of L-NNA and MRS2500 enhanced Ca²⁺ transients (Fig. 4G; Appendix Table S4). However, this excitatory effect was strongly diminished in the presence of secretin (Fig. 4H). During

EFS the excitatory response was diminished in the presence of secretin (100 nM) and few Ca²⁺ transients were rescued after EFS. In the presence of secretin, the frequency of Ca²⁺ transients rescued post EFS was 3.36 ± 1.49 (Ca²⁺ transients/10 s) (Fig. 4I), the area of Ca²⁺ events was 0.96 ± 0.39 (μm*s) (Fig. 4J), duration of events was 1.26 ± 0.46 (ms) (Fig. 4K) and spatial spread was 5.9 ± 2.1 (μm) (Fig. 4L; Appendix Table S4). These results indicate that secretin has the ability to override the Ca²⁺ generating effects of excitatory neurotransmission in ICC-DMP indicating an axis between the excitatory neurotransmission pathway and secretin modulation of ICC-DMP.

## Secretin dampens responses to exogenous cholinergic stimulation

ICC-DMP express receptors and mechanisms for transducing enteric motor neurotransmitter signaling. However, due to the expression of muscarinic receptors by smooth muscle cells (Chen et al, 2007; Lee et al, 2017), it is possible that a portion of the whole-muscle responses to EFS is mediated by direct stimulation of these

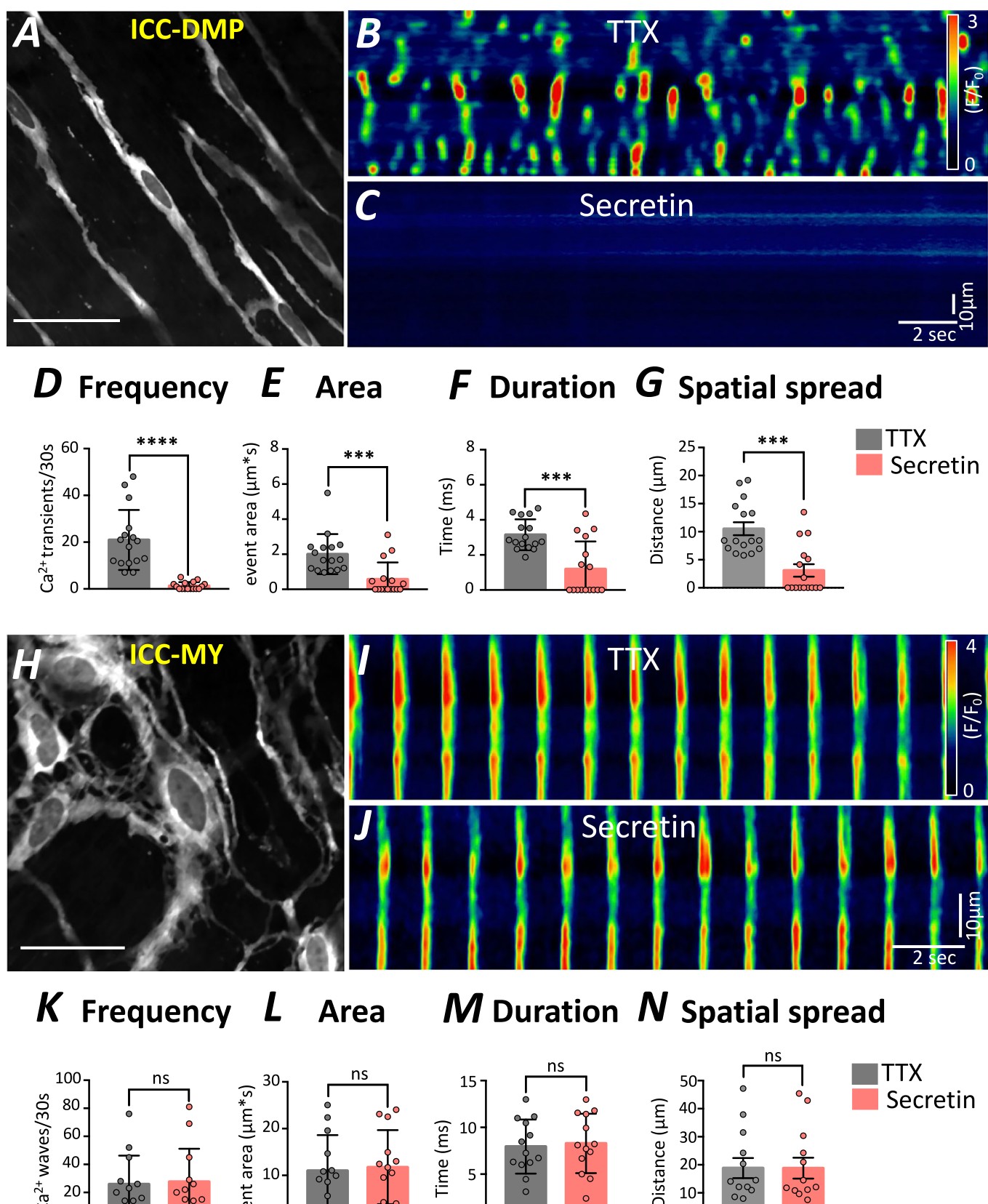

**Figure 3.  Secretin inhibited Ca²⁺ transients in ICC-DMP but not in ICC-MY.**

Ca²⁺ signaling was recorded from ICC in small intestinal muscles from Kit-iCre-GCaMP6f mice. (A) ICC-DMP in the deep muscular plexus near the mucosal surface of the circular muscle. Scale bar represents 40 μm. (B) Ca²⁺ transients in ICC-DMP plotted as STMaps in the presence of TTX (1 μM) and (C) after the addition of secretin (100 nM). All parameters of Ca²⁺ transients were reduced by secretin, (D) frequency (per 30 s), (E) area of events (μm*s), (F) duration of events (ms), and (G) spatial spread of Ca²⁺ transients (μm). (H) Ca²⁺ transients in ICC-MY, between the circular and longitudinal muscle layers in plane of the myenteric plexus of the small intestine, were unaffected by secretin (100 nM) (I, J). Ca²⁺ events in ICC-MY appeared as waves of activity spreading through networks of cells, as previously reported (Drumm et al, 2017). These wave-like events appear as continuous bands of Ca²⁺ activity in STMaps and did not show significant quantitative changes in the presence of secretin. (K) frequency (per 30 s), (L) area of events (μm*s), (M) duration of events (ms), and (N) spatial spread of Ca²⁺ transients (μm). The data are plotted as mean ± SEM, and significance was determined using paired *t* test. ****$P < 0.0001$ (D); ***$P = 0.0001$ (E); ***$=P = 0.0001$ (F); ***$P = 0.0002$ (G); cells = 16, $n = 8$ for ICC-DMP and cells = 13, $n = 6$ for ICC-MY preparations. Source data are available online for this figure.

SMCs. This subsequently modulate Ca²⁺ sensitization of the contractile apparatus (Gao et al, 2016; Kitazawa and Somlyo, 1990; Somlyo and Somlyo, 2003) or influence membrane potential and excitability (Inoue and Isenberg, 1990; Tsvilovskyy et al, 2009). Perhaps the effects of secretin could be overwhelmed by these downstream regulatory mechanisms. Acetylcholine is a major excitatory neurotransmitter in GI muscles, so we tested the effects of exogenous carbachol, an analogous muscarinic agonist, on muscle contractions in the presence of secretin. Secretin reduced the force of contractions of jejunal muscles pre-stimulated with carbachol (CCh; 10 μM) (Fig. 5A). Contractile AUC was reduced to 60.2% ± 5.79 (mN*min) (Fig. 5B) and amplitude of contractions reduced to 61.1% ± 4.90 (mN) (Fig. 5C) after the addition of secretin. Contractile tone (baseline) was also reduced to 88.5% ± 2.47 (mN) after the addition of secretin (Fig. 5D), but the frequency of contractions was unaffected ($P = 0.89$; Fig. 5E). We also compared the maximum force of contraction to CCh (10 μM) under control conditions (Fig. 5F, brown line) and in the presence of secretin (Fig. 5F, pink line). The AUC and amplitude of contraction were significantly reduced by secretin pretreatment (Fig. 5G,H). In the presence of secretin (100 nM), the maximum contraction to CCh AUC was reduced by 74.7% (mN*min) (Fig. 5G) and amplitude of contractions was reduced by 58.3% (mN) (Fig. 5H). These data show that secretin inhibits contractile responses to whole-muscle excitatory stimulation via muscarinic mechanisms.

## Secretin inhibits IP3-mediated Ca²⁺ release in ICC-DMP

Experiments above suggest that secretin modulates the contractility of small intestinal muscles, an effect that could slow postprandial intestinal motility to allow for optimized acid neutralization in the proximal small intestine. The next part of the study sought to understand cellular regulation imposed by secretin by examining the mechanism by which secretin inhibits Ca²⁺ transients in ICC-DMP. Regulation of Ca²⁺ transients in ICC-DMP occurs, in part, by Gαq-linked muscarinic (M3) and neurokinin A (NKA) receptors stimulation of IP3 production (Baker et al, 2018b; Zhu et al, 2015). We used a combination of cell-directed photo-uncaging of IP3, using a Micropoint laser system (10 pulses,10 sec;10% intensity power; Andor, Oxford Instruments) in combination with imaging of Ca²⁺ transients in ICC-DMP to determine whether Ca²⁺ release from IP3 receptors is impeded by secretin (Fig. 6A). Under control conditions, Ca²⁺ transients in ICC-DMP were greatly enhanced by uncaging of IP3 (Fig. 6B) and this effect was significantly inhibited by secretin (Fig. 6C). Uncaging of IP3 increased Ca²⁺ transients

firing frequency to 30.3 ± 5.55 (per 30 s) (Fig. 6D), increased the area of events to 3.09 ± 0.67 (μm*s) (Fig. 6E), as well as the duration to 4.14 ± 0.62 (ms) (Fig. 6F). In the presence of secretin (100 nM), uncaging of IP3 failed to elicit significant responses in ICC-DMP (Fig. 6D–G; Appendix Table S5). These findings show that secretin blocks Ca²⁺ release from IP3 receptors in ICC-DMP and suggests a mechanism for the inhibitory effects on whole-muscle contractile function in the presence of secretin.

## Secretin mediates its inhibitory effects via Ca²⁺ release mechanisms

We also sought to determine if Ca²⁺ influx mechanisms are involved in secretin responses by testing the effects of removal of Ca²⁺ from the external solution using a nominal-Ca²⁺ Krebs solution (Fig. 7). For these experiments, a brief puff of secretin was used to capture changes in Ca²⁺ handling under conditions in which Ca²⁺ was omitted from the Krebs solution (nominally Ca²⁺ free). As before in controls secretin inhibited Ca²⁺ transients in ICC-DMP with external Ca²⁺ at 2 mM (Fig. 7A,B). While the frequency of events was diminished in the presence secretin, few events remained in two preparations and Ca²⁺ transient frequency was reduced to 4.67 ± 0.04 (per 30 s) (Fig. 7E), area of events was reduced to 0.64 ± 0.45 (μm*s) (Fig. 7F), however, event duration and spatial spread remained unchanged (Fig. 7G,H; Appendix Table S6). Application of secretin (100 nM) in the presence of nominal external Ca²⁺ also inhibited Ca²⁺ transients (Fig. 7C,D). In nominal-Ca²⁺ Krebs, secretin inhibited Ca²⁺ transient frequency to 0.60 ± 0.40 (per 30 s) (Fig. 7E) and reduced Ca²⁺ event area to 0.88 ± 0.55 (μm*s) (Fig. 7F). No significant changes to the duration of Ca²⁺ transients (Fig. 7G) or spatial spread (Fig. 7H) were observed. Control puffs of Krebs had no effect on Ca²⁺ transients in ICC-DMP (Fig. 7I,J). Additionally, secretin did not have any significant effect in ex vivo preparations pretreated with the SERCA pump inhibitor, thapsigargin (10 μM) (Fig. EV3A–H). These experiments indicate that the Ca²⁺ inhibitory effects visualized in ICC-DMP by secretin are due to Ca²⁺ release mechanisms and not by influx mechanisms.

## Activation of GPCR-Gαs in Kit-positive cells mimic the activity of secretin

Secretin receptors are coupled to effects via Gαs (FK et al, 2006). We bred floxed Designer Receptors Exclusively Activated by Designer Drugs (DREADDs) GPCR-Gs mice (Akhmedov et al, 2017) with Kit-iCre mice to express Gs-DREADD in ICC.

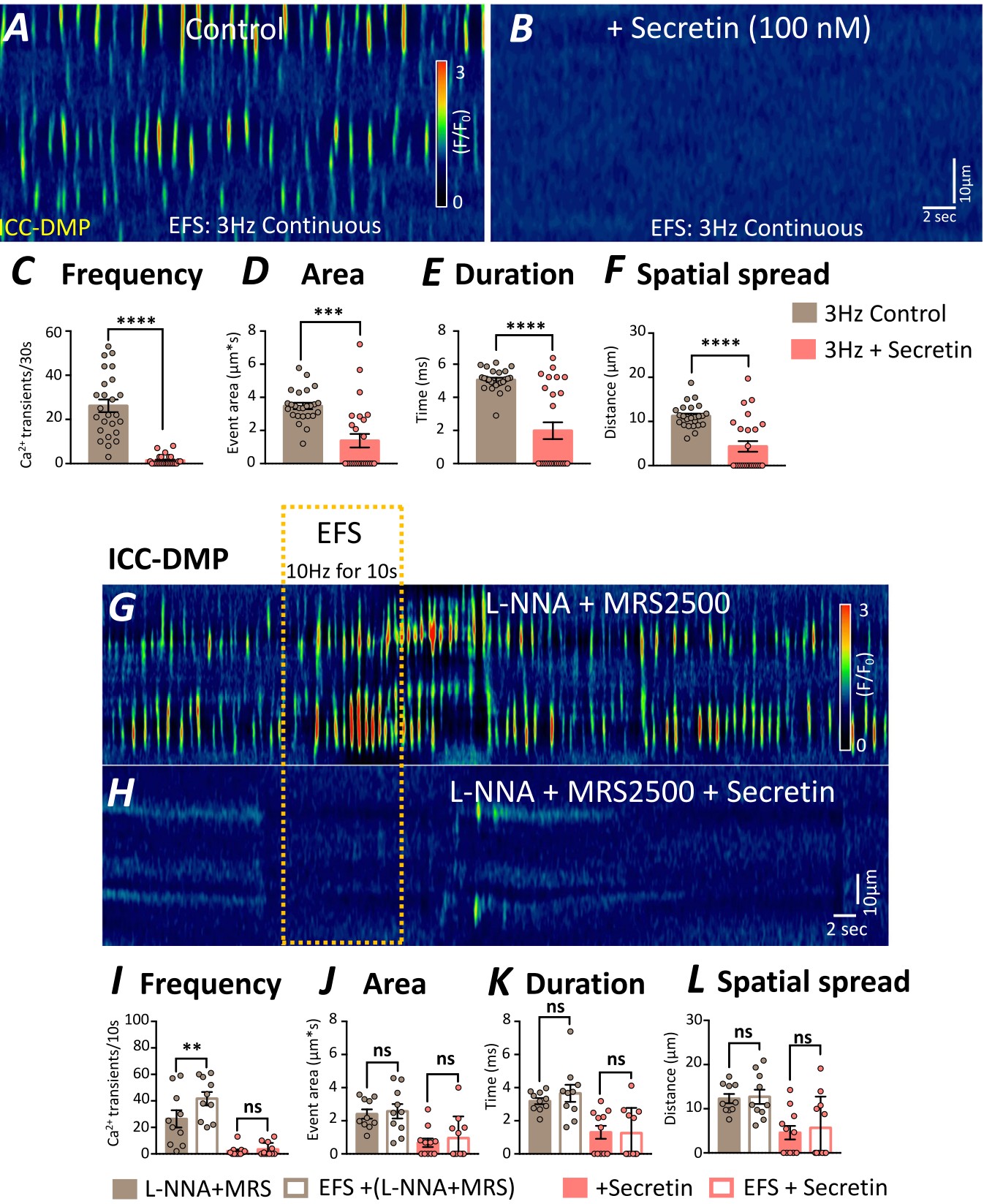

Figure 4. Secretin dampened responses to excitatory neurotransmission in ICC-DMP.

(A) Ca$^{2+}$ transients were recorded from ICC-DMP during continuous electrical field stimulation (EFS, 3 Hz, 15 s period) and the Ca$^{2+}$ transients were plotted as STMaps. (B) In the presence of secretin (100 nM), all Ca$^{2+}$ transient parameters in ICC-DMP were reduced (C–F). (C) frequency (per 30 s), (D) area of events (µm*s), (E) duration of events (ms), and (F) spatial spread (µm). (G) Shows the effects of EFS (10 Hz for 10 s) on Ca$^{2+}$ transients in the presence L-NNA (100 µM) and MRS2500 (1 µM) to block contributions from enteric inhibitory neurons. EFS increased the frequency of Ca$^{2+}$ transients. (H) Shows the effects of secretin (100 nM) on responses to EFS. Scale bars in H apply to STMaps in (G). (I–L) Summary of effects of EFS on Ca$^{2+}$ transients before and after addition of secretin: (I) frequency of Ca$^{2+}$ transients (per 10 s), (J) area of events (µm*s), (K) duration of events (ms), and (L) spatial spread of Ca$^{2+}$ transients (µm). Note that excitatory EFS was unable to rescue Ca$^{2+}$ release in the presence of EFS. The data are plotted as mean ± SEM, and significance was determined using paired $t$ test. ****$P < 0.0001$ (C); ***$P = 0.0002$ (D); ***$P < 0.0001$ (E); ****$P < 0.0001$ (F); cells = 25, $n = 3$ for 3 Hz EFS and cells = 12, $n = 5$ for 10 Hz EFS. Source data are available online for this figure.

Immunohistochemistry of muscles from these mice showed that Gs-DREADD (green; Fig. 8A) was exclusive to c-Kit-positive cells (red; Fig. 8B); colocalization was confirmed in a merged image (Fig. 8C). Higher magnification (60x objective) confirmed expression and colocalization of DREADD receptors in ICC-DMP (Fig. 8D) and ICC-MY (Fig. 8E).

In jejunal muscles of Kit-iCre-Gs-DREADD mice the DREADD activator, clozapine-N-oxide (CNO) (10 µM) inhibited contractions in a manner similar to the effects of secretin (Fig. 8F–H). The AUC and amplitude of contractions were reduced significantly by CNO to 56.0% ± 6.76 (mN*min) and 67.3% ± 4.82 (mN), respectively (Fig. 8I,J; Appendix Table S7), but the frequency of contractions was unaffected ($P = 0.58$) (Fig. 8K). These results indicate that activation of the Gs-DREADD receptor in ICC and subsequent activation of cAMP causes decreases in AUC and amplitude but not frequency in small intestinal muscle strip experiments.

## cAMP analogs attenuate the force of contraction in murine small intestine

A major consequence of liberation of Gαs is activation of adenylate cyclase and generation of cAMP (Ostrom et al, 2022). Thus, the downstream mechanism of secretin may result from enhanced cAMP levels due to its receptor coupling through Gαs. Therefore, we tested the effects of a canonical means of activating adenylyl cyclase and generating cAMP in cells (Fujino et al, 2010) by testing whether forskolin mimicked the effects of secretin activation on small intestinal contractions. Forskolin (20 nM) reduced the amplitude and AUC of contractile events in small intestine muscles (Fig. 9A–F). The AUC was reduced to 38.3% ± 5.93 (mN*min) (Fig. 9D) and amplitude of contractions was reduced to 51.6% ± 5.60 (mN) (Fig. 9E). Forskolin at this low concentration had no significant effect on frequency ($P = 0.16$) (Fig. 9F; Appendix Table S8). We also tested the effects of a membrane permeable cAMP analog. 8-Bromo-cAMP (500 µM) also reduced the AUC and amplitude of contractions without effects on frequency (Fig. 9G–L; Appendix Table S8). 8-Bromo-cAMP inhibited jejunal contractions, AUC was reduced to 21.2% ± 5.24 (mN*min) (Fig. 9J) and contractile amplitude was reduced to 36.7% ± 7.15 (mN) (Fig. 9K). The frequency of contractions was not significantly affected by 8-Bromo-cAMP ($P = 0.39$) (Fig. 9L). These results show that elevation of cAMP had very similar effects as secretin in terms of modulating intestinal contractions.

## 8-Bromo-cAMP inhibits calcium transients in ICC-DMP

We next sought to determine whether Ca$^{2+}$ signaling in ICC-DMP is also modulated by the 8-Bromo-cAMP. Ca$^{2+}$ transients were monitored in ICC-DMP before (Fig. 10A) and after addition of 8-Bromo-cAMP (500 µM) (Fig. 10B). Ca$^{2+}$ transients were significantly reduced by 8-Bromo-cAMP to 11.3 ± 2.94 (per 30 s) (Fig. 10C). The other Ca$^{2+}$ transient parameters were not reduced significantly: area of Ca$^{2+}$ events 1.10 ± 0.21 (µm*s) (Fig. 10D), duration 2.57 ± 0.41 (ms) (Fig. 10E) and spatial spread 6.3 ± 1.1 (µm) (Fig. 10F). The effects of 8-Bromo-cAMP (500 µM) were also tested on the rhythmic Ca$^{2+}$ transients of ICC-MY (Fig. 10G–L), but this compound had no significant effect on Ca$^{2+}$ transients in ICC-MY (Fig. 10I,L), frequency ($P = 0.73$), Ca$^{2+}$ event area ($P = 0.56$), duration ($P = 0.85$) or spatial spread ($P = 0.98$) (Fig. 10I–L; Appendix Table S9).

## Measuring cAMP in ICC-DMP in response to secretin

Changes in cAMP levels in ICC-DMP were directly evaluated using FRET EPAC reporter mice (CAMPER) crossed with Kit-iCre mice to express the cAMP sensor exclusively in ICC. Immunohistochemistry experiments on jejunal muscles from CAMPER-Kit-iCre mice (Fig. 11A–F) showed that the cAMP sensor (green; Fig. 11B,E) was expressed in c-Kit-positive cells (red; Fig. 11A,D) as shown in merged images (Fig. 11C,F). Levels of cAMP were examined in response to either forskolin, secretin or a DMSO vehicle (Fig. 11G–L). Forskolin and secretin perfusions showed significant increases in the FRET ratio (mTurquoise/Venus) signals and the FRET efficiency was significantly increased to 115% ± 2.27 in response to forskolin (1 µM) (Fig. 11G,H) and to 108% ± 0.87 in response to secretin (100 nM) (Fig. 11I,J). DMSO controls had no effect on FRET ($P = 0.27$) (Fig. 11K,L; Appendix Table S10). This data confirms that secretin increases cAMP level in ICC-DMP.

## Secretin mediates its effects via PKA-dependent mechanism

Multiple downstream signaling pathways can be activated by cAMP, including cAMP-dependent protein, protein kinase A (PKA) and exchange protein directly activated by cAMP (EPAC) (Xiaodong et al, 2008). We evaluated the effects of the PKA antagonist (AT7867; 10 µM) on jejunal muscle strips contractions (Fig. 12A–G). As above, secretin (100 nM) inhibited the AUC and amplitude of jejunal muscle contractions (Fig. 12A,B; Appendix Table S11). However, this effect was significantly attenuated in the presence of the PKA antagonist (AT7867; 10 µM; 20 min) (Fig. 12C,D). Secretin alone reduced AUC to 49.6% ± 6.22 (mN*min) (Fig. 12E), amplitude to 62.6% ± 6.18 (Fig. 12F), and had no effect on frequency of contractions ($P = 0.62$) (Fig. 12G). When AT7867 was present, the effects of secretin were reduced (Fig. 12E,F).

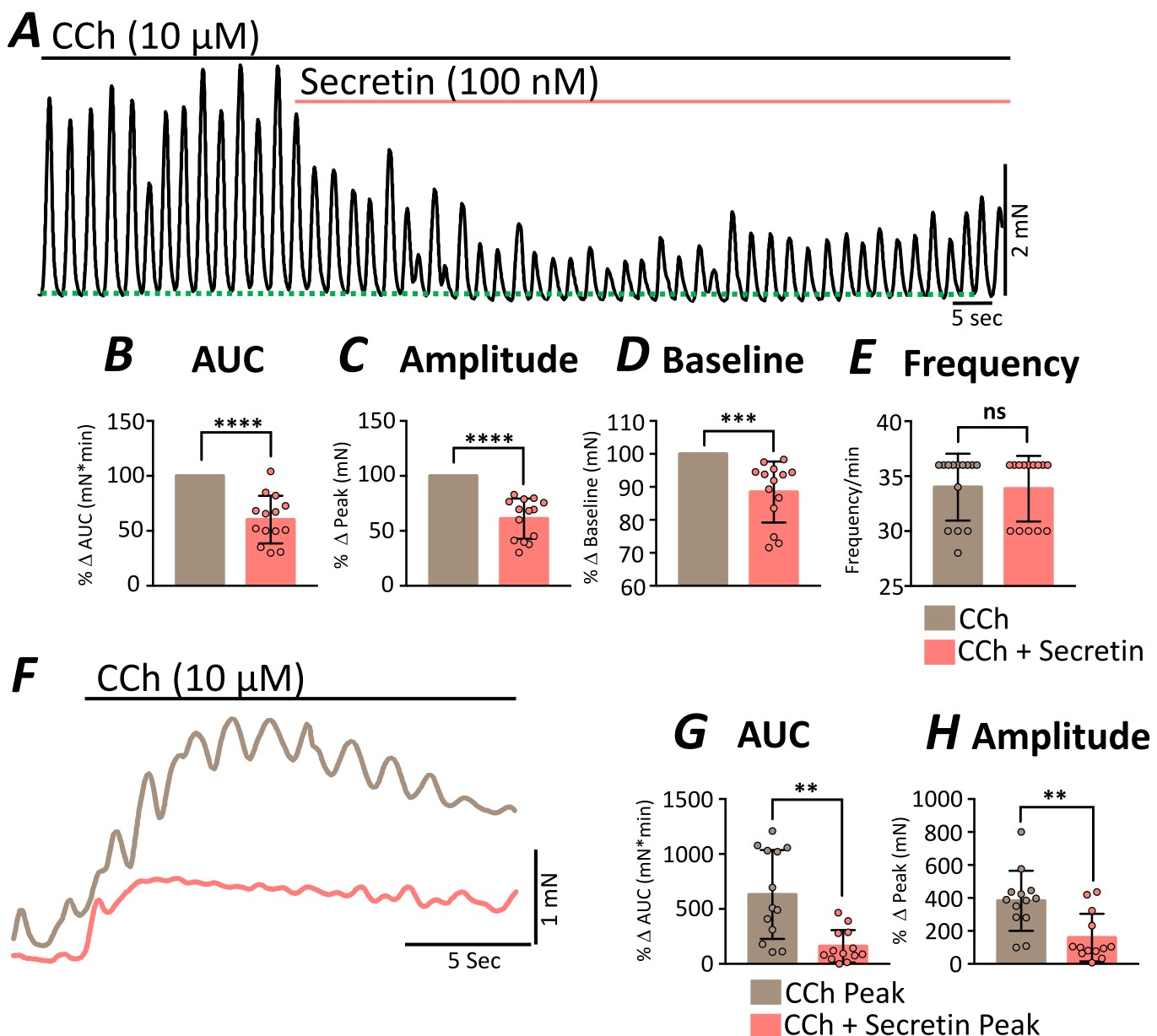

**Figure 5. Secretin dampened whole-muscle contractile responses to muscarinic stimulation.**

(A) Small intestinal muscles were stimulated with carbachol (CCh; 10 μM), and during continuous exposure to CCh were exposed to secretin (100 nM). (B) The area under the curve (AUC; mN*min) and (C) amplitude of contractions (mN) were decreased by secretin. (D) The baseline of contractions (mN) was also reduced as indicated by the dotted green line. (E) Contractile frequency was unchanged (min⁻¹). (F) Maximum peak muscle contractions in response CCh (brown trace) and with secretin (100 nM, pink trace) pretreatment. Both AUC (G) and maximum amplitude (H) of contractions were reduced. All data were normalized to controls and data were plotted as percent change from control. The data are plotted as mean ± SEM, and significance was determined using paired $t$ test. ****$P < 0.0001$ (B); ****$P < 0.0001$ (C); ***$P = 0.0004$ (D); ns = 0.8903 (E); **$P = 0.0018$ (G); **$P = 0.0036$ (H); strips = 14, $n = 5$. Source data are available online for this figure.

The PKA antagonist also inhibited the effects of secretin on $Ca^{2+}$ signaling in ICC-DMP (Fig. 12H–M). In this series of experiments, secretin alone inhibited ICC-DMP $Ca^{2+}$ transient frequency to $3.12 \pm 1.36$ from $41.7 \pm 3.61$ ($Ca^{2+}$ events per 30 s) (Fig. 12J), area of $Ca^{2+}$ events $0.53 \pm 0.17$ from $1.68 \pm 0.17$ (μm*s), (Fig. 12K), duration $1.33 \pm 0.42$ from $3.48 \pm 0.20$ (ms), (Fig. 12L) and spatial spread $2.6 \pm 0.75$ from $7.3 \pm 0.6$ (μm) (Fig. 12M).

In the presence of AT7867, secretin (100 nM) did not inhibit $Ca^{2+}$ transients in ICC-DMP (Fig. 12H,I). $Ca^{2+}$ transient frequency was slightly reduced $11.9 \pm 2.81$ compared to $14.9 \pm 2.90$ (per 30 s) (Fig. 12J). All other $Ca^{2+}$ transient parameters were not changed, area of $Ca^{2+}$ events ($P = 0.61$) (Fig. 12K), duration ($P = 0.10$) (Fig. 12L) and spatial spread, ($P = 0.45$) (Fig. 12M; Appendix Table S12).

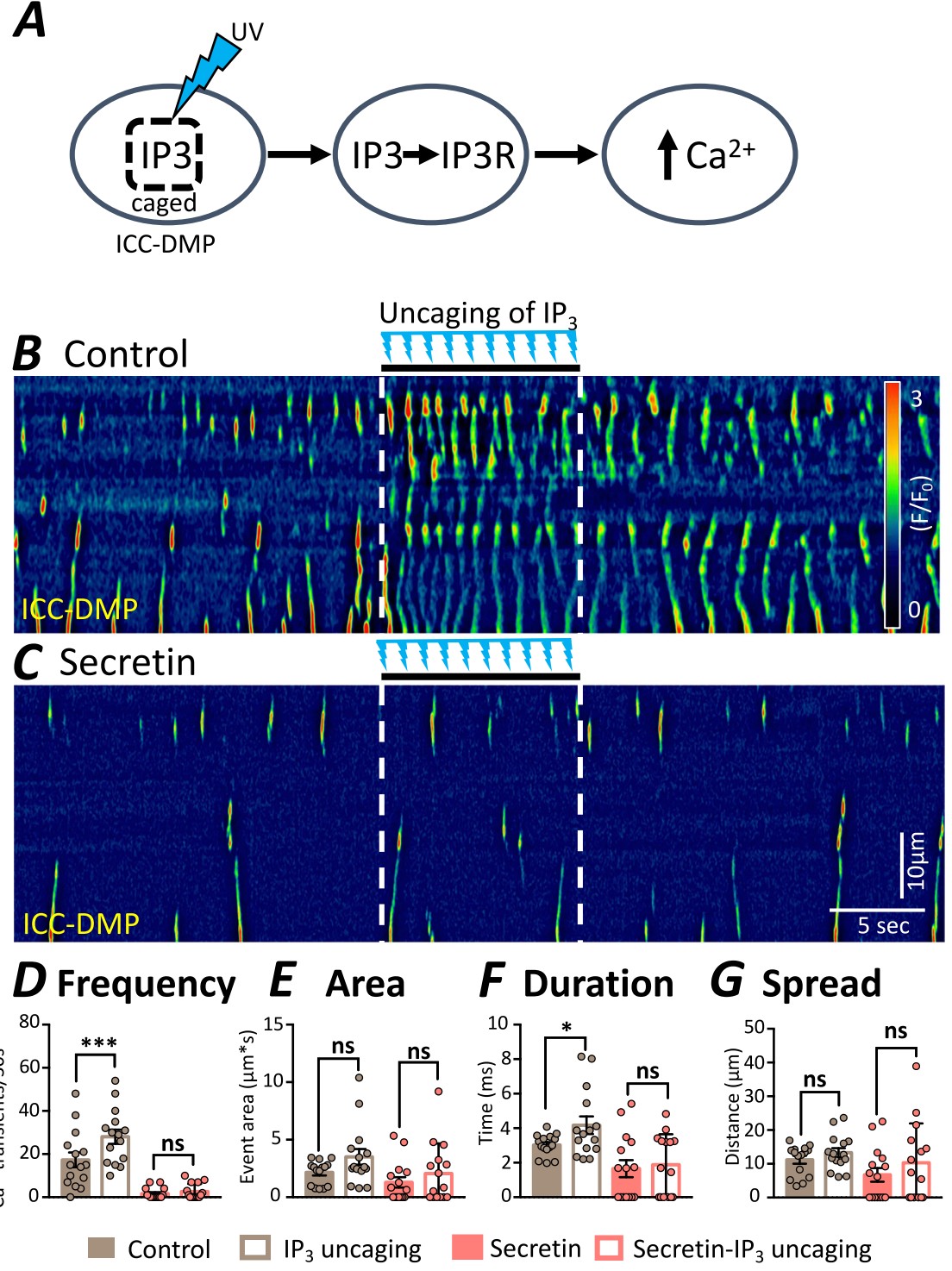

**Figure 6. Secretin inhibits IP3-dependent Ca²⁺ release from IP3Rs in ICC-DMP.**

(A) Schematic depicting photo-uncaging of IP3 and initiating a rise in Ca²⁺ in ICC-DMP. (B) Ca²⁺ transients in ICC-DMP plotted as STMaps before, during and after photo-uncaging of IP3 with UV light (10 pulses for 10 s). Note: Photo-uncaging of IP3 increased Ca²⁺ transients. (C) Photo-uncaging of IP3 in the presence of secretin (100 nM) failed to increase Ca²⁺ transients in ICC-DMP. (D–G) Summary graphs of Ca²⁺ transient parameters before and after IP3-uncaging before (brown) and in the presence of secretin (pink): (D) frequency of Ca²⁺ transients (per 30 s), (E) area of events (µm*s), (F) duration of events (ms), and (G) spatial spread of Ca²⁺ transients (µm). The data are plotted as mean ± SEM, and significance was determined using paired $t$ test. ***$P = 0.0002$ (D); *$P = 0.0420$ (F); cells = 10, $n = 4$. Source data are available online for this figure.

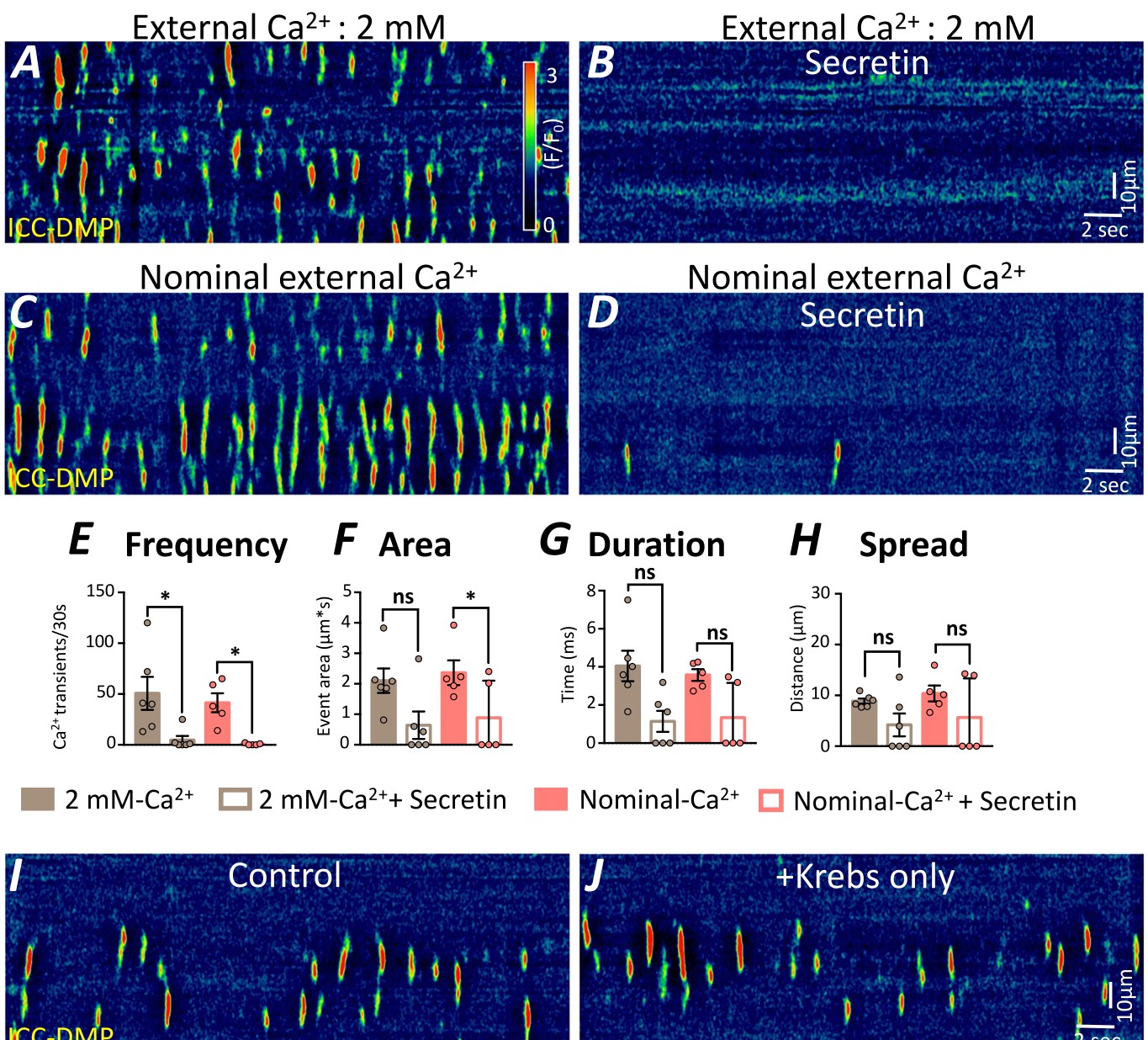

**Figure 7. Inhibition of Ca²⁺ transients in ICC-DMP was unaffected by nominal Ca²⁺ gradient.**

(A) Ca²⁺ transients in ICC-DMP plotted as STMaps under control condition with 2 mM-Ca²⁺ in the external solution. (B) Secretin (100 nM) inhibited Ca²⁺ transients with 2 mM-Ca²⁺ in the external solution. (C) Under nominal external Ca²⁺ conditions (D) secretin inhibited Ca²⁺ transients under nominal external Ca²⁺ conditions. Summary graphs of Ca²⁺ transient parameters: (E) frequency of Ca²⁺ transients (per 30 s), (F) area of events (μm*s), (G) duration of events (ms), and (H) spatial spread of Ca²⁺ transients (μm) were plotted for the effects of secretin in 2 mM-Ca²⁺ external solution (brown) and nominal Ca²⁺ external solution (Pink). (I) Ca²⁺ STMap of ICC-DMP in Krebs solution and (J) after addition of warmed Krebs with the same external Ca²⁺ concentration. The data are plotted as mean ± SEM, and significance was determined using paired *t* test. *$P = 0.0445$ (E); *$P = 0.0111$ (F); cells = 7, $n = 5$. Source data are available online for this figure.

A small effect of secretin on contractions persisted in the presence of the PKA antagonist. Thus, it is possible that additional effects of the secretin could be mediated by another effector, such as EPAC. In the presence of the EPAC inhibitor, ESI-05 (10 μM), the inhibitory effects of secretin were slightly attenuated but still showed significant reductions in amplitude and AUC of jejunal contractions (Fig. 13A–G). The AUC reduced to 66.5% ± 4.61 (Fig. 13E) and amplitude to 82.5% ± 4.19

(Fig. 13F) but not frequency of events ($P = 0.51$) (Fig. 13G; Appendix Table S13).

ESI-05 failed to reverse the inhibitory effects of secretin on ICC-DMP Ca²⁺ transients (Fig. 13H–M), as secretin was still shown to inhibit Ca²⁺ transients in ICC-DMP in the presence of ESI-05. Ca²⁺ transient frequency was reduced to 3.27 ± 1.70 compared to 16.0 ± 2.43 (per 30 s) (Fig. 13J). All other Ca²⁺ transient parameters were also significantly reduced: area of Ca²⁺ events 0.56 ± 0.24

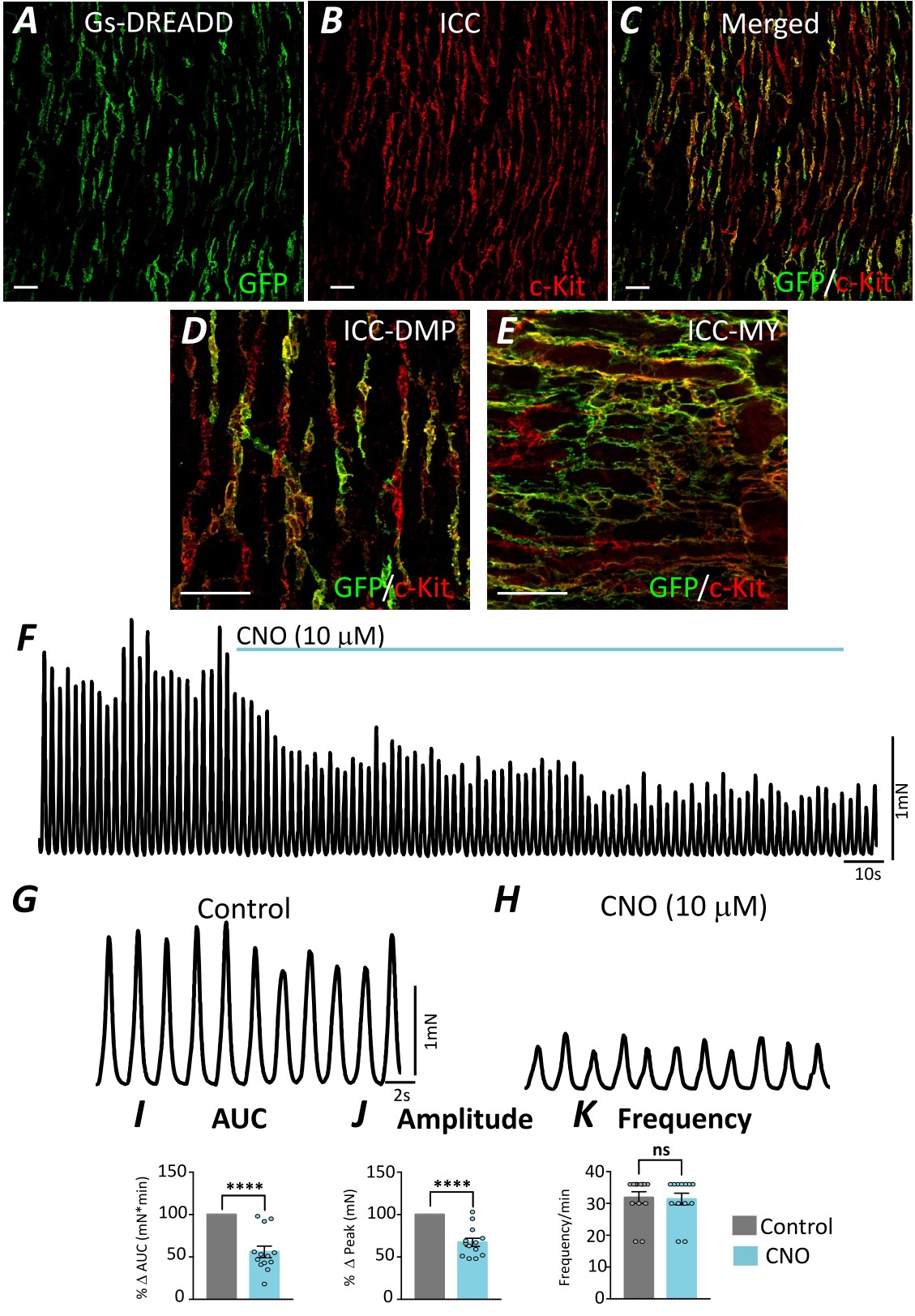

**Figure 8. Chemogenetic activation of Gs affected pathways in ICC.**

Jejunal muscles from the Kit-iCre-Gs-DREADD mouse were utilized and expression Gs-DREADD in ICC was confirmed by immunohistochemistry. (A) Green fluorescent protein (GFP) antibodies for expression of DREADD, (B) ICC antibodies of c-Kit (red) and (C) merge of Gs-DREADD and ICC images. Scale bar is 50 µm for all images A-C using 20x objective. (D) Higher magnification images (60x objective) showing merged images of Gs-DREADD and c-Kit expression (GFP/c-Kit) in ICC-DMP and (E) in ICC-MY. Scale bar is 50 µm for images in (D, E). (F) Contractions of jejunal muscles from Kit-iCre-Gs-DREADD mice before and after addition of clozapine-N-oxide (CNO; 10 µM) to liberate Gs-DREADD in ICC. (G) Expanded trace from the control region in (F, H) during the response to CNO (10 µM). (I) The area under the curve (AUC; mN*min) and (J) the amplitude of contractions (mN) were reduced. However, (K) the frequency (per 30 s) of contractions was not affected. The data, except contraction frequency, were normalized to control levels. The data are plotted as mean ± SEM, and significance was determined using paired $t$ test. ****$P < 0.0001$ (I, J); Strips = 13, $n = 5$. Source data are available online for this figure.

compared to $2.69 \pm 0.36$ (µm*s) (Fig. 13K), duration $1.03 \pm 0.38$ compared to $3.24 \pm 0.18$ (ms) (Fig. 13L) and spatial spread $3.7 \pm 1.5$ compared to $13.2 \pm 1.5$ (µm) (Fig. 13M; Appendix Table S14).

## Discussion

Secretin is a GI hormone and is known to influence gastrointestinal (GI) motility (Brandler et al, 2020; Lu and Owyang, 1995, 2009), however it's mechanism(s) of action for how it affects motility remained unclear. In this study, we show that transcripts of secretin receptors (Sctr) are highly expressed by ICC-DMP of the small intestine and binding of secretin inhibits $Ca^{2+}$ signaling in ICC-DMP and reduces the force of contractions. Interestingly, the frequency of contractions were unaffected by secretin which is consistent with the lack of receptor expression by ICC-MY, the pacemaker cells in the small intestine (Huizinga et al, 1995; Ward et al, 1994). SCTR is a class-B GPCR that couples to cellular effects predominantly through Gαs to activate adenylate cyclase and increase cAMP production and, in some circumstances, it couples to Gq to mobilize intracellular $Ca^{2+}$ (Garcia et al, 2012). In ICC-DMP SCTR coupling appears to be through Gαs and cAMP production. Secretin directly enhanced cAMP levels in ICC-DMP, as observed in the FRET-based cAMP sensor mice (CAMPER), and effects of secretin were mimicked by a membrane permeable analog of cAMP. The effects of secretin on contractions were also simulated in muscles of Gs-DREADD mice with expression of the receptors directed to ICC by application of CNO. Antagonists of PKA and EPAC showed that secretin-dependent effects were initiated via activation of PKA, and only minor effects might be attributed to EPAC. Finally, $Ca^{2+}$ transients, which are due to release of $Ca^{2+}$ from IP3R in ICC-DMP (Baker et al, 2016), were strongly stimulated by uncaging of IP3 and this effect was inhibited by secretin.

Regulation of $Ca^{2+}$ signaling through cAMP and PKA-dependent phosphorylation of IP3R is well-established (Taylor, 2017/05) (Nakade et al, 1994; Tang et al, 2003; Wagner et al, 2008; Wagner et al, 2004; Wagner et al, 2003; Wojcikiewicz and Luo, 1998). In many cells generation of cAMP is coupled to mechanisms that stimulate $Ca^{2+}$ release, as seen in cardiac pacemakers (Berisha et al, 2021; Capel et al, 2021). Results from the present study suggest a mechanism in which activation of PKA leads to inhibition of $Ca^{2+}$ release. Previous studies have reported inhibition of IP3R via cAMP and PKA in megakaryocytes (Tertyshnikova and Fein, 1998), airway smooth muscle (Bai and Sanderson, 2006), pancreatic acinar cells (Giovannucci et al, 2000a; Giovannucci et al, 2000b), and vascular smooth muscle cells (Cuíñas et al, 2016). However, the precise mechanistic steps for these responses is still unknown. Our data suggest that a similar phenomenon occurs in ICC-DMP, aligning with these observations, as initiation of $Ca^{2+}$ transients by

uncaging of IP3 was inhibited by secretin, suggesting this effect is mediated by inhibition of IP3R. Alternative mechanisms might include cAMP-PKA or EPAC-mediated inhibition of store-operated calcium entry (SOCE), as previously suggested (Zhang et al, 2019). However, our experiments tend to rule out a role for SOCE, as the effects of secretin persisted after reduction in the transmembrane $Ca^{2+}$ gradient. Another potential mechanism is PKA-mediated $Ca^{2+}$ clearance from the cytosol via phosphorylation of phospholamban and activation of the SERCA pump (Schwinger et al, 1998; Tada et al, 1998). However, excessive $Ca^{2+}$ accumulation in the endoplasmic reticulum might lead to enhanced $Ca^{2+}$ release through IP3R (Aschar-Sobbi et al, 2012).

Previous studies have suggested that secretin slows GI transit by targeting vagal afferent pathways originating from the gastroduodenal mucosa (Lu and Owyang, 1995). From gene expression studies (Lee et al, 2017) and confirmed by RNAscope in the present study, we identified abundant expression of the secretin receptor (Sctr) in ICC-DMP in the small intestine. A recent study has also reported expression of secretin receptors in human enteric neurons (Drokhlyansky et al, 2020), suggesting possible involvement of enteric inhibition neurons in the effects of secretin. However, we found that the inhibitory effects of secretin on intestinal contractions persisted in the presence of TTX. These findings indicate that the inhibitory effects of secretin are independent of regulation by enteric motor neurons and suggests that responses are due to direct effects on post-junctional cells of the SIP syncytium, namely ICC-DMP.

Regulation of $Ca^{2+}$ transients in ICC-DMP is a major transduction mechanism for excitatory and inhibitory motor neurotransmitters in the small intestine (Baker et al, 2018a; Baker et al, 2018b). $Ca^{2+}$ transients activate ANO1 channels in ICC, generating inward current that conducts to other cells of the SIP syncytium via electrical coupling. In the case of SMCs, enhancing inward current causes depolarization and increases the open probability of L-type $Ca^{2+}$ channels, $Ca^{2+}$ entry and contraction. Excitatory neurotransmitters enhance $Ca^{2+}$ transients in ICC-DMP (Baker et al, 2018b), and conversely, the inhibitory neurotransmitter NO, suppresses $Ca^{2+}$ transients in these cells (Baker et al, 2018a). Our findings indicate that secretin regulation also affects $Ca^{2+}$ release in ICC-DMP, suggesting that it may alter or even interfere with neural responses. This may explain why, under certain experimental conditions, the effects of secretin appeared to be mediated by enteric neurons in some studies. For example, under the influence of secretin we found that the normally stimulatory effects of enteric excitatory motor neurotransmission on $Ca^{2+}$ release in ICC-DMP was greatly diminished. $Ca^{2+}$ transients in ICC-DMP and contractile force of muscles was also not rescued by whole-muscle stimulation by CCh, even with

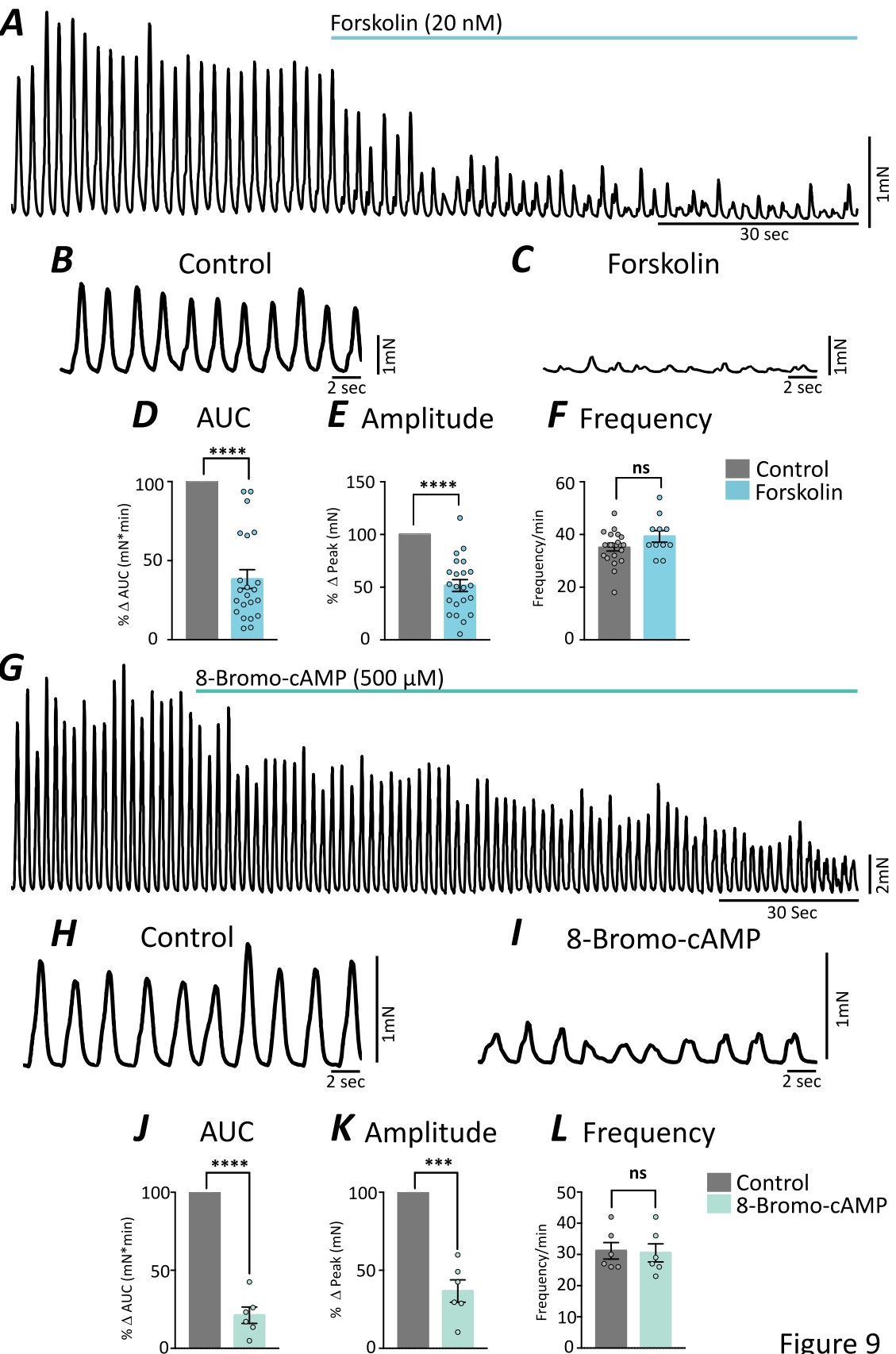

**Figure 9**

**Figure 9. Forskolin and cAMP analog inhibit small intestinal contractions.**

(A) Jejunal muscle strip contractions were reduced by the adenylate cyclase activator, forskolin (20 nM) in a manner similar to the effects of secretin. (B) Expanded trace during control period and (C) during response to forskolin. (D) The area under the curve (AUC; mN*min) and (E) the amplitude of contractions (mN) were reduced by forskolin, but (F) contractile frequency (per 30 s) was unaffected. (G) cAMP analog, 8-Bromo-cAMP (500 µM) reduced muscle contractions. (H) Expanded trace showing contractions during the control period and (I) during the response to 8-Bromo-cAMP. (J) Contraction AUC and (K) amplitude of contractions were reduced by forskolin, but (L) the frequency of contractions was unaffected. Data were normalized to control levels, except contraction frequency. The data are plotted as mean ± SEM, and significance was determined using paired $t$ test. ****$P < 0.0001$ (D, E, J); ***$P = 0.0003$ (K); 8-Bromo-cAMP, strips = 6, $n = 4$ and Forskolin, strips = 22, $n = 8$. Source data are available online for this figure.

inclusion of the excitatory effects of direct muscarinic stimulation of SMCs (Gao et al, 2016; Inoue and Isenberg, 1990; Kitazawa and Somlyo, 1990; Somlyo and Somlyo, 2003; Tsvilovskyy et al, 2009). In ICC-DMP excitatory neurotransmitters, ACh and neurokinins, coupled to responses via Gαq-mediated enhancement in IP3 production. Our experiments using photo-uncaging of IP3 showed that $Ca^{2+}$ release from IP3R is less probable in the presence of secretin. Thus, the effectiveness of regulatory agonists utilizing IP3 and IP3R in ICC-DMP to enhance motor output is diminished. Taken together these results suggest the following scheme for regulation of small intestinal motility by secretin: (1) Secretin receptors are abundantly expressed by ICC-DMP and are activated by secretin binding. (2) Coupling of SCTR via Gαs leads to enhanced cAMP generation and activation of PKA. (3) By a mechanism not yet understood, PKA reduces $Ca^{2+}$ release from IP3R. (4) Inhibition of $Ca^{2+}$ release in ICC-DMP reduces inward currents caused by activation of ANO1 channels (Zhu et al, 2015). (5) Reduction of net inward currents in the SIP syncytium leads to reduction in the force of small intestinal contractions.

This study demonstrates a novel role for secretin in regulation of small intestinal motility. The effects of secretin are mediated through coupling of SCTR via Gαs, enhanced cAMP production and activation of PKA. The link between PKA and activation of IP3R has not been identified but our experiments suggest a new pathway. The steps of this pathway are schematized in Fig. 14. This study demonstrates a new concept: ICC-DMP serve as a focal point upon which neural and hormonal signaling converge, yielding integration of signaling for regulation of intestinal motility. The effects of secretin on $Ca^{2+}$ transients in ICC-DMP represent a novel and important pathway for hormonal regulation of intestinal motility during the phases of digestion. This mechanism may serve to slow intestinal transit, ensuring sufficient time for digestion and absorption of nutrients as well as adequate neutralization of acidic chyme exiting the stomach.

## Methods

**Reagents and tools table**

| Reagent or resource | Source | Identifier |
| --- | --- | --- |
| **Experimental models: organisms/strains** | | |
| GCaMP6f-floxed mice (B6;129S-Gt(ROSA) 26Sor$^{tm95.1(CAG-GCaMP6f)Hze}$/J) | Jackson Laboratories | Strain #024105 RRID: IMSR_JAX:024105 |
| Kit-Cre mice (c-Kit$^{+/Cre-ERT2}$) | Provided by Dr. Dieter Saur, Technical University Munich, Munich, Germany | https://doi.org/10.1113/JP271699 |

| Reagent or resource | Source | Identifier |
| --- | --- | --- |
| C57BL/6-Gt(ROSA) 26Sor$^{tm1(CAG-ECFP*/Rapgef3/Venus*)Kama}$/J | Jackson Laboratories | Strain #032205 |
| **Software and algorithms** | | |
| Fiji, version 2.0.0-rc-69/1.52 | NIH | https://fiji.sc/ |
| 4SM, Subcellular Fluorescence Analysis Software | GitHub | https://doi.org/10.1016/j.isci.2022.104277 https://github.com/SharifAmit/CalciumGAN/tree/main |
| STMapAuto, $Ca^{2+}$ Analysis plugin | GitHub | https://doi.org/10.1016/j.ceca.2020.102260 https://github.com/gdelvalle99/STMapAuto |
| **Drugs and reagents** | | |
| Secretin, human | GenScript | Cat. No. RP10249 |
| 8-Bromo-cAMP | Tocris Bioscience | Cat. No. 1140 |
| MRS 2500 tetraammonium salt | Tocris Bioscience | Cat. No. 2159 |
| Nicardipine hydrochloride | Sigma-Aldrich | Cat. No. N7510 |
| Apamin | Sigma-Aldrich | Cat. No. A9459 |
| N$_\omega$-Nitro-L-arginine | Sigma-Aldrich | Cat. No. N5501 |
| Tetrodotoxin (citrate) | Cayman Chemical Company | Cat. No. 14964 |
| Caged inositol triphosphate (ci-IP3/PM) | Tocris Bioscience | Cat. No. 6210 |
| RNAscope™ Multiomic C1 Channel Reagents | ACDBio | Cat. No. 322935 |
| RNAscope™ Multiplex Fluorescent Reagent Kit v2: | ACDBio | Cat. No. 323100 |
| RNAscope Multiplex Fluorescent Detection Kit | ACDBio | PN 323110 |
| RNAscope H202 and Protease Reagents | ACDBio | PN 322381 |
| RNAscope Target Retrieval reagents | ACDBio | PN 322000 |

| Reagent or resource | Source | Identifier |
|---|---|---|
| RNAscope Wash Buffer | ACDBio | PN 310091 |
| TSA buffer | ACDBio | PN 322809 |
| RNAscope™ Probe- Mm-Sctr | ACDBio | Cat. No. 508511 |
| RNAscope 3-plex Negative control Probe -Mm | ACDBio | Cat. No. 320871 |
| RNAscope™ 3-plex Positive Control Probe- Mm | ACDBio | Cat. No. 320881 |
| co-detection antibody dilutant | ACDBio | Cat. No. 323160 |
| Opal 570 | Akoya Biosciences | PN FP1488001KT |
| Vectashield Vibrance Antifade Mounting Medium with DAPI | Vector laboratories | Cat. No. H-1800 |
| Antibodies | | |
| Goat polyclonal primary antibody- mSCFR | R and D systems | Cat. No. AF1356 |
| Donkey anti-goat Alexa Fluor® 488 | Jackson ImmunoResearch Laboratories Inc. | Cat. No. 705-545-147 |

## Animals

The animals used and the experiments performed in this study were in accordance with the National Institutes of Health Guide for the Care and Use of Laboratory Animals. All procedures were approved by the Institutional Animal Use and Care Committee at the University of Nevada, Reno (IACUC Protocols ID: 24-06-1216, 21-02-1131-1). GCaMP6-floxed mice (B6.129S-Gt (ROSA)26Sortm38(CAG-GCaMP6)Hze/J), cAMP Encoded Reporter, CAMPER mice (C57BL/6-Gt (ROSA)26Sor$^{tm1(CAG-ECFP*/Rapgef3/Venus*)}$ $^{Kama}$/J)) and associated wild-type siblings (C57BL/6) were purchased from The Jackson Laboratory. The Gs-DEARDD mice were kindly donated by Dr. Rebecca Berdeaux (UTHealth, Houston, TX, USA). (Kit-Cre mice (c-Kit + /Cre-ERT2) were gifted from Dr. Saur (Technical University Munich, Munich, Germany). Once crossed, 6–8 week old Kit-Cre-GCaMP6f, or Kit-iCre-Gs-DREADD mice were injected with 2 mg tamoxifen for three consecutive days as described previously (Baker et al, 2016) to induce cre recombinase activation and subsequent GCaMP6f, CAMPER or Gs-DREADD expression and were anaesthetized by inhalation with isoflurane (Baxter) and killed by cervical dislocation.

Cynomolgus monkeys (2–8 years, 2.5–7.5 kg) were housed at Charles River Laboratories, Reno, NV. All procedures followed NIH Guide for the Care and Use of Laboratory Animals and were approved by IACUC (#I-001358). Euthanasia involved ketamine sedation (10 mg/kg), Beuthanasia-D (0.7 mL), and exsanguination. No animals were bred or euthanized solely for this study.

## Muscle preparation

Segments of the jejunal small intestine, approximately 1–2 cm in length, were removed from mice after an abdominal incision, and tissues were bathed in Krebs-Ringer bicarbonate solution (Krebs). The tissues were opened along the mesenteric border, and intra-luminal contents were washed away with cold Krebs. The mucosa and submucosa layers were then removed by sharp dissection and muscles were pinned flat in a 50 mm Sylgard-coated dish with the serosal side of the muscles facing the bottom of the dish.

## Drugs and solutions

All muscles were perfused and maintained with Krebs solution containing (mmol/liter): 120.35 NaCl, 15.5 NaHCO$_3$, 5.9 KCl, 1.2 MgCl$_2$, 1.2 NaH$_2$PO$_4$, 2.5 CaCl$_2$, and 11.5 glucose. Krebs was warmed to a physiological temperature of $37 \pm 0.2$ °C and diffused with a mixture of 97% O$_2$–3% CO$_2$. In one set of experiments, a Krebs solution was used in which Ca$^{2+}$ omitted (nominally Ca$^{2+}$ free solution). Secretin was purchased from GenScript (Cat. No. RP10249), 8-Bromo-cAMP (Cat. No. 1140) and MRS 2500 tetraammonium salt (Cat. No. 2159) were purchased from Tocris Bioscience; nicardipine hydrochloride (Cat. No. N7510), and N$_\omega$-Nitro-L-arginine L-NNA (Cat. No. N5501) were purchased from Sigma-Aldrich; Tetrodotoxin (citrate) was purchased from the Cayman Chemical Company, Ann Arbor, Michigan, US, (Cat. No. 14964),

## RNAscope tissue preparation

Cross sections of muscle and flat mounts were prepared from wild-type (C57BL/6) jejunum. For cross sections, intact tissue segments were pinned to the base of a Sylgard-coated dish. A capillary tube of suitable size was inserted to ensure that the lumen remained open during fixation. For flat sections, tissue segments were opened along the mesenteric border and pinned flat. The mucosa was then peeled away to leave the isolated muscle layer. Muscles were fixed with 4% paraformaldehyde (PFA) for 24 h and washed for 24 h in 1X azide-free phosphate-buffered saline (PBS). Next, muscles were dehydrated in increasing concentrations of sucrose in 1xPBS (5%, 10%, and 15%, for 30 min each, and then 20% overnight) at 4 °C. Muscles were then embedded in a 1:1 ratio of Tissue Tek embedding medium for frozen tissue specimens (Sakura Finetek USA, Inc., Torrance, CA) and 20% sucrose. Intact segments for cross-sections were placed standing upright in embedding blocks, and muscle layer sheets for flat sections were placed circular side down at the bottom of the blocks. Blocks were frozen at -80 °C and kept until sectioning. Sections (10–12 μm thick) were cut with a Leica CM3050 S cryostat (Leica, Nussloch, Germany). For flat sections, every section was collected on a slide and examined under a dissection microscope until the first pieces of tissue were observed in 10–12 μm section increments. The next few sections from this point were collected to guarantee that the ICC-DMP layer would be processed.

## RNAscope in situ hybridization and immunofluorescence

The RNAscope system was used according to the manufacturer's directions (Advanced Cell Diagnostics Bio, Newark, CA) and as described previously in (Aresta Branco et al, 2023). The RNAscope Intro Pack for Multiplex Fluorescent Reagent Kit v2- Mm (ACDBio, Cat. No. 323100) was utilized to permeabilize tissues, hybridize probes and amplify the probe signal as detailed in the ACDBio RNAscope Multiplex Fluorescent Reagent Kit V2 sample preparation user manual (UM No. 323100, ACDBio). Included in the kit

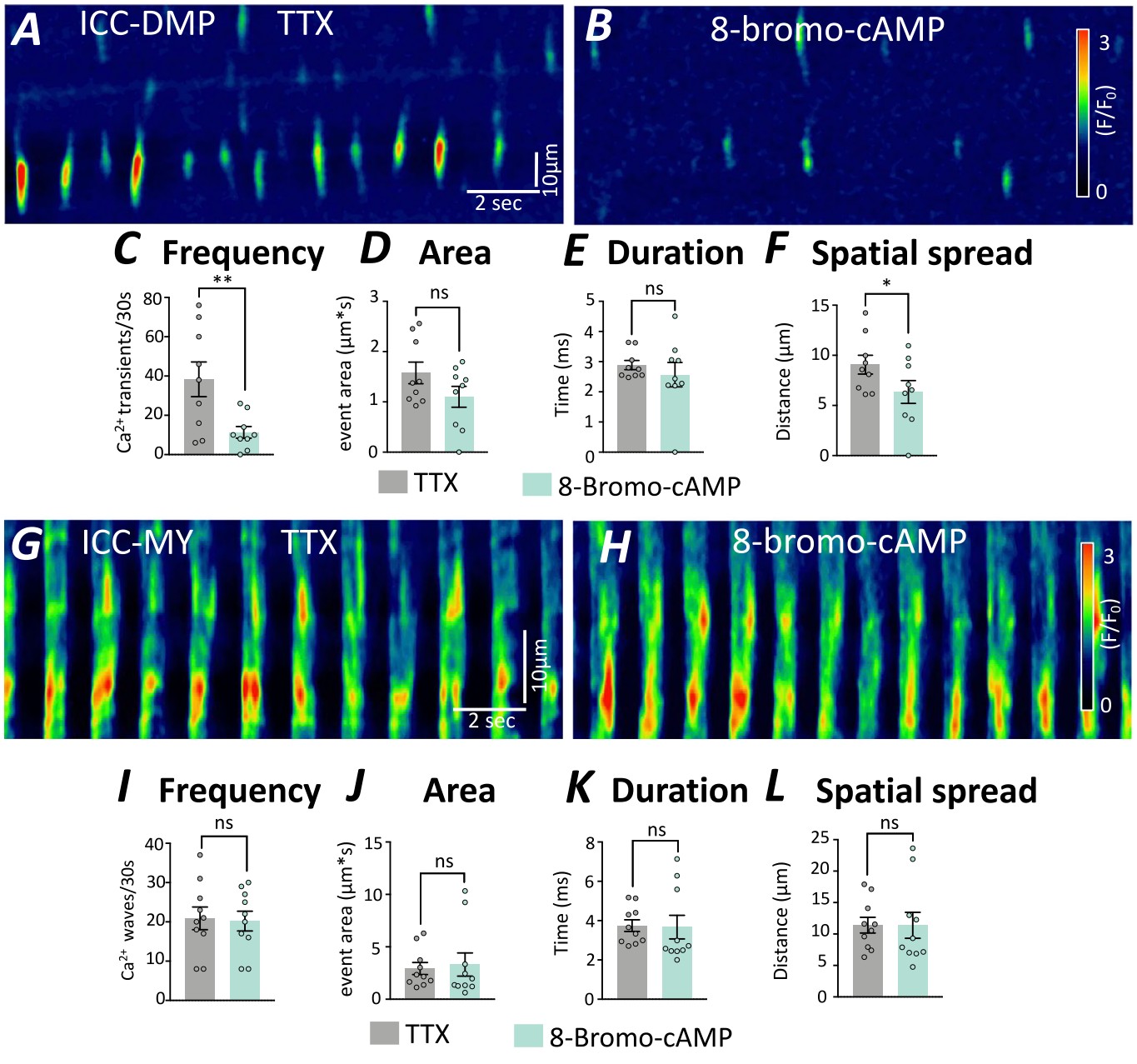

**Figure 10.  cAMP analog inhibited Ca²⁺ transients in ICC-DMP.**

Ca²⁺ transients were recorded from ICC in small intestinal muscles of Kit-iCre-GCaMP6f mice. (A) Ca²⁺ signals of ICC-DMP were plotted as STMaps before and (B) after the addition of 8-Bromo-cAMP (500 μM). Ca²⁺ transient parameters were inhibited by 8-Bromo-cAMP application, (C) frequency of Ca²⁺ transients (per 30 s), (D) area of events (μm*s), (E) duration of events (ms), and (F) spatial spread of Ca²⁺ transients (μm). (G, H) Ca²⁺ waves in ICC-MY of small intestine were unaffected by 8-Bromo-cAMP (500 μM). None of the Ca²⁺ transient parameters of ICC-MY were changed: (I) frequency of Ca²⁺ waves (per 30 s), (J) area of events (μm*s), (K) duration of events (ms), and (L) spatial spread of Ca²⁺ transients (μm). The data are plotted as mean ± SEM, and significance was determined using paired *t* test. **$P = 0.0032$ (**C**); *$P = 0.0383$ (**F**). For ICC-MY cells $= 10$, $n = 3$ and for ICC-DMP cells $= 9$, $n = 4$. Source data are available online for this figure.

was as follows: RNAscope Multiplex Fluorescent Detection Kit (PN 323110), RNAscope H202 and Protease Reagents (PN 322381, ACDBio) and RNAscope Target Retrieval reagents (PN 322000, ACDBio), RNAscope Wash Buffer (PN 310091, ACDBio), TSA buffer (322809, ACDBio), and detection reagents for manual amplification. The Opal fluorophore from Akoya Biosciences was purchased separately: Opal 570 (PN FP1488001KT, Akoya

Biosciences). The protocol was performed as follows: All prepared slides were washed with freshly made 1× azide-free PBS and pH adjusted to 7.2 for 5-min and baked in a dry hybridizer for 30-min at 60 °C (Dako Hybridizer - Cat. No. S2450). Tissues were post-fixed with pre-chilled 4% PFA for 15 min at 4 °C, step-dehydrated with 50%, 70%, and 100% (x2) ethanol for 5 min each at room temperature (RT) and air dried for at least 5 min. Next, RNAscope

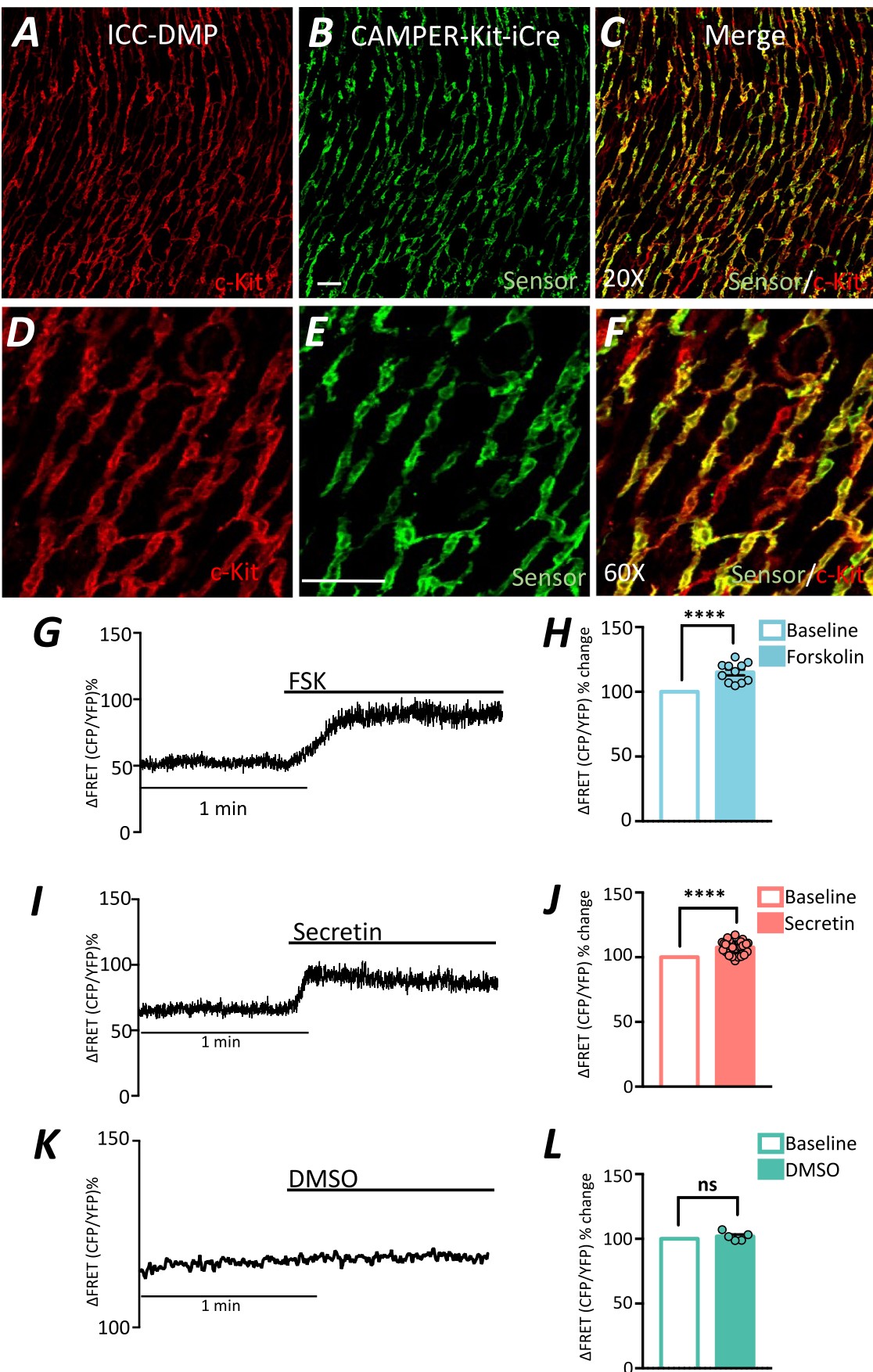

**Figure 11.  Secretin increase cAMP levels in ICC-DMP.**

(A–F) Whole-mounts of jejunal muscles from CAMPER-Kit-iCre mouse were immuno-stained for (A) c-Kit (red) and (B) the cAMP sensor (mTurquoise/Venus emission signal, color coded with green). (C) Merged images of (A, B) showing that the cAMP sensor is localized in ICC (×20 objective). (D) Higher magnification images (×60 objective) of ICC-DMP, (E) the cAMP sensor, and (F) a merged image of D&E. Scale bars represent 50 µm. (G, H) FRET ratio (mTurquoise/Venus) in ICC-DMP before and after addition of forskolin (1 µM). (I, J) FRET ratio in ICC-DMP before and after addition of secretin (100 nM). (K, L) FRET ratio in ICC-DMP before and after addition of DMSO vehicle. Data are summarized as percent change in FRET ratio (normalized to 100% of controls). The data are plotted as mean ± SEM, and significance was determined using paired $t$ test. ****$P < 0.0001$ (H, J); *ns$=P = 0.3664$ (L); DMSO, cells = 5, secretin, cells = 30, $n = 3$ and forskolin, cells = 11, $n = 3$. Source data are available online for this figure.

Hydrogen Peroxide, (PN 322381, ACDBio) was applied (10-min, RT), then washed with distilled water (x2) for 2-min each. Slides were immersed in 1× Co-detection Target Retrieval solution (Cat. No. 323165, ACDBio) (heated to 98–102 °C in a steamer) for 5 min, then washed in distilled water (x2) for 2-min each and washed with 1XPBS-Tween (0.1%) (PBS-T) (x2) for 2-min each. A working solution of primary antibody in co-detection antibody dilutant (Cat. No. 323160, ACDBio) was prepared at a 3× concentration per antibody and applied to each slide and incubated at 4 °C overnight. On day-2, slides were washed with PBS-T (×2) for 2 min each and then post-fixed in cold 4% PFA for 30 min at RT and then washed with PBS-T (×2) for 2 min each again. RNAscope Protease plus Protease Plus reagent (PN 322381, ACDBio) was applied for 30 min at 40 °C in a RapidFISH Slide Hybridizer (Cat. No. 240200, Boekel Scientific, Feasterville, PA). The next step was to hybridize the probe of interest: Sctr RNAscope Probe (Cat. No. 508511) was used for detection of secretin receptor RNA (2 h, 40 °C). For assay validation and result interpretation the Positive Control Probe was used: POLR2A (Channel C1), (RNAscope Positive Control Probe) Cat. No. 320881) as well as the negative control probe to detect the bacterial dapB gene and to control for background noise (RNAscope Negative Control Probe, Cat. No. 320871). Slides were subsequently washed twice with 1× RNAscope® Multiplex BSA Wash Buffer (Cat. No. 310091 and three amplification steps were then performed using the RNAscope Multiplex Fluorescent Reagent Kit (Cat. No. 323110), washing with wash buffer (x2) for 2 min each in-between each step: AMP 1 (30 min, 40 °C), AMP 2 (30 min, 40 °C), and AMP 3 (15 min, 40 °C). The horseradish peroxidase (HRP)-C1 from the RNAscope™ Multiomic C1 Channel Reagents (Cat. No. 322935, ACDBio) was then applied to develop the HRP signal (15-min, 40 °C) and subsequently washed with wash buffer (x2) for 2 min each. After washing, Opal 570 1:1000 (Cat. No. FP1488001KT, Akoya Biosciences, Marlborough, MA) diluted in TSA buffer from the RNAscope™ Multiomic C1 Channel Reagents (Cat. No. 322935, ACDBio), was applied (30-min, 40 °C) and washed with wash buffer (x2) for 2 min each. RNAscope HRP blocker was then applied (15 min, 40 °C) and washed with wash buffer (×2) for 2 min each and then slides were washed with PBS-T (×2) for 2 min each. Secondary antibodies were prepared in co-detection antibody dilutant (Cat. No. 323160, ACDBio) and incubated for 30-min at RT. Slides were washed with PBS-T (×2) for 2 min each and then mounted. Nuclear counterstain was achieved with DAPI included in the mounting media, Vectashield Vibrance Antifade Mounting Medium with DAPI (Cat. No. H-1800, Vector Laboratories, Newark, CA). Confocal images were taken at ×20 and ×60 magnification on a Nikon A1R HD25 confocal (Nikon Instruments Inc., Melville, NY).

All images were analyzed using Fiji/ImageJ software (Schindelin et al, 2012). The ACDBio semi-quantitative guidelines for assay interpretation were utilized based on the recently detailed system as previously published (Aresta Branco et al, 2023) and can be found at https://acdbio.com/dataanalysisguide. Score guidelines are as follows: score 0: no staining or <1 dot/10 cells; score 1: 1–3 dots/cell; score 2: 4–9 dots/cell, no or very few dot clusters; score 3: 10–15 dots/cell and/or <10% of dots are in clusters; score 4: >15 dots/cell and/or >10% of dots are in clusters. Briefly, the RNAscope signal is indicated by fluorescent punctate. Clusters may also be present which are multiple mRNA molecules overlapping or are in very close proximity to each other. The total probe count for each cluster was determined by methods where each cluster area was divided by the area of a single probe as previously described (Aresta Branco et al, 2023), and in this instance each probe was (0.62 pixel Width × 0.62 pixel Height). The average number of punctate were then calculated per cell per region of interest for ICC-DMP and ICC-MY and a subsequent ACDBio score was assigned for each cell type. To simplify data interpretation, plots were made for Sctr RNAscope-positive ICC cells depicted as percent per region of interest (256 pixels × 256 pixels) for both ICC-DMP and ICC-MY of distinct ROI's at ×20 and ×60 magnification. An unpaired Student's $t$ test was performed comparing column A (ICC-DMP) Sctr-positive ICC and column B (ICC-MY) Sctr-positive ICC. Positive and negative controls were used to control for background noise and images were analyzed as previously described (Aresta Branco et al, 2023).

## Muscle strip contractility measurements

Jejunal muscles (2 cm in length and ~3 mm in diameter) were pinned down in a Sylgard-lined dish and opened longitudinally. Mucosa and submucosa were peeled away by sharp dissection leaving the tunica muscularis intact. Muscle strips cut parallel to the circular axis were attached to a fixed mount and the opposite end to an isometric strain gauge (Fort 10; Radnoti, ADinstruments, Colorado Springs, CO, USA). Muscle strips were equilibrated for 45 min in 37 °C Kreb's buffer with 4 mN of passive tone applied. Contractility was recorded using AcqKnowledge software (3.9.1; Biopac Systems, Goleta, CA).

## Calcium imaging

Flat prep small intestinal tissues were pinned in imaging chambers, perfused with warmed Krebs at 37 °C and equilibrated for 1 h. Preparations were then visualized and imaged with a spinning-disk confocal system (CSU-W1; spinning disk, Yokogawa Electric, Tokyo, Japan) attached to an upright Nikon Eclipse FN1 microscope was used for Ca²⁺ imaging. The system was equipped with two solid-state laser lines of 488 nm and 561 nm. The laser

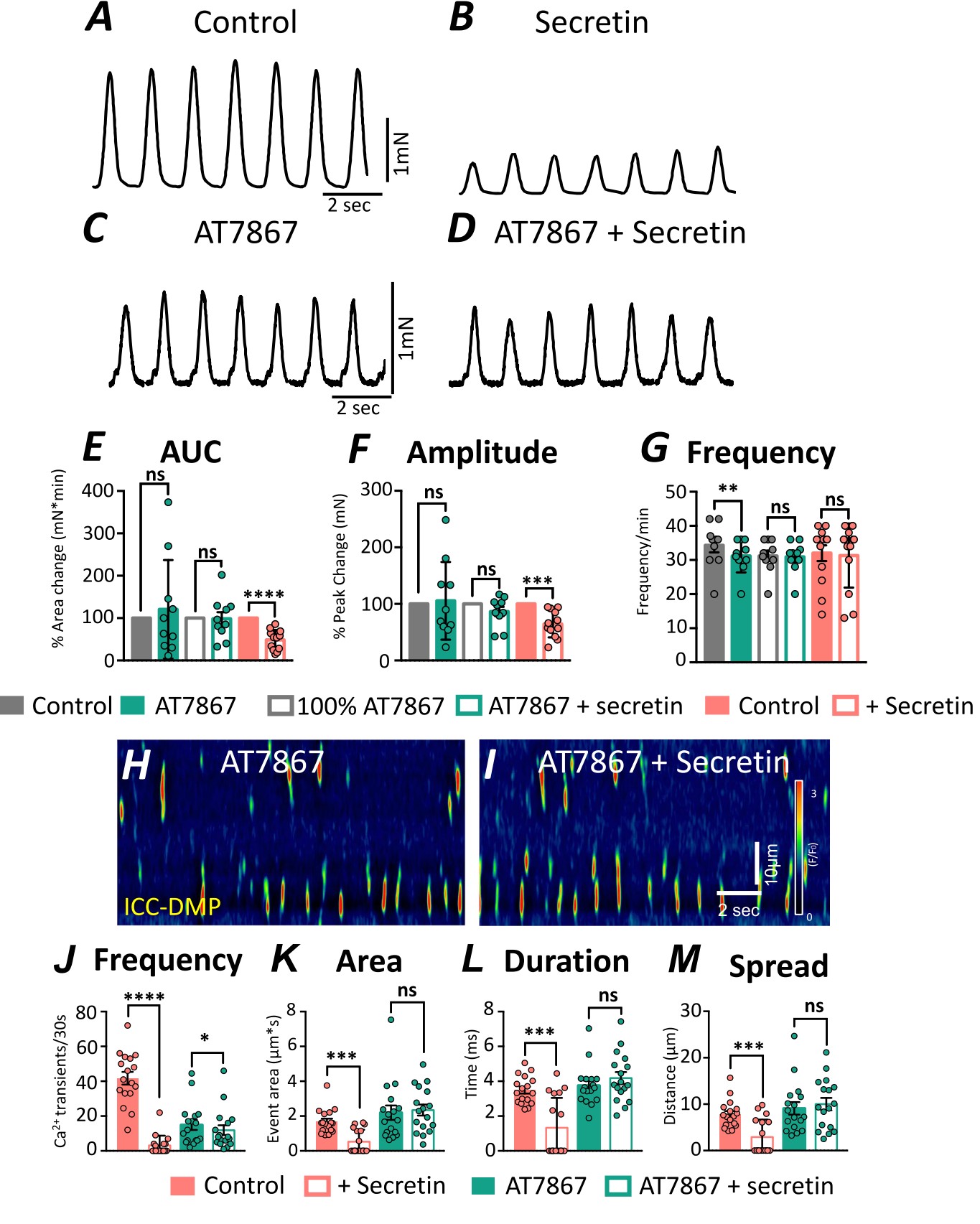

**Figure 12. Effects of secretin are reduced by PKA antagonist.**

(A, B) Jejunal muscle contractions were inhibited by secretin (100 nM). (C, D) The effects of secretin were reduced in muscles pretreated with PKA antagonist, AT7867 (10 μM). (E–G) Summary of the effects of secretin with and without AT7867: (E) The inhibitory effects of secretin on the area under the curve (AUC; mN*min) of contractions and (F) amplitude (mN) of contractions were reduced by AT7867, but (G) the frequency (min$^{-1}$) of contractions was unaffected. The effects of secretin on Ca$^{2+}$ transients were also reduced by the PKA antagonist. (H) shows Ca$^{2+}$ transients in ICC-DMP plotted as STMaps in the presence of AT7867 (10 μM) and (I) after addition of secretin (100 nM). (J–M) Summary data showing that Ca$^{2+}$ transient parameters were unchanged by secretin in the presence of AT7867: (J) frequency of Ca$^{2+}$ transients (per 30 s), (K) area of events (μm*s), (L) duration of events (ms), and (M) spatial spread of Ca$^{2+}$ transients (μm). The data are plotted as mean ± SEM, and significance was determined using paired $t$ test. ****$P < 0.0001$ (E); ***$P = 0.0001$ (F); **$P = 0.0036$ (G); ****$P < 0.0001$; *$P = 0.0312$ (J); ***$P = 0.0006$ (K); ***$P = 0.0007$ (L); ***$P = 0.0005$ (M). For contraction experiments: secretin alone strips = 12, $n = 5$ and AT7867+secretin, strip = 10, $n = 4$. For Ca$^{2+}$ measurement experiments cells = 19, $n = 12$. Source data are available online for this figure.

lines are combined with a Borealis system (ANDOR Technology, Oxford Instruments). Two high-speed electron multiplying charged coupled devices (EMCCD) cameras (Andor iXon-Ultra 897 EMCCD Cameras; ANDOR Technology, Oxford Instruments) are mounted to the system to maintain fast speed acquisition at full frame of 512 × 512 active pixels, as previously described(Baker et al, 2021). Image sequences were acquired using water immersion Nikon CFI Fluor lenses (10 × 0.3 NA, 20 × 0.5 NA, 40 × 0.8 NA, 60 × 0.8 NA and 100 × 1.1 NA) (Nikon Instruments, New York, USA) at 33–50 fps and MetaMorph software (MetaMorph Inc., TN, USA), as previously described (Baker et al, 2015). In brief, images were captured using a high-speed EMCCD Camera (Andor iXon Ultra; ANDOR Technology). Image sequences were collected at 33–105 fps using MetaMorph software (MetaMorph Inc.). All experiments were performed in the presence of nicardipine (100 nM) to reduce movement artifacts. In experiments with pharmacological agents, a control activity period (30 s) was recorded, and then solutions containing the drugs were perfused into the chamber and recorded. In zero-Ca$^{2+}$ imaging experiments, a warmed (37 °C), 500 μL secretin (100 nM) solution in Zero-Ca$^{2+}$ Krebs buffer was puffed into the imaging dish during the recording to visualize the immediate Ca$^{2+}$ shutdown mechanism response to secretin. A control recording was taken before the puff and control puffs containing normal warmed Krebs were also taken to confirm the drug effect.

## Analysis of Ca$^{2+}$ imaging experiments

Differentiation between ICC-MY and ICC-DMP was based on the following. (a) Their stereotypical anatomical localizations with muscles: ICC-DMP are located near the bulk of the circular muscle layer, and ICC-MY are located between the longitudinal and circular muscle layers (Komuro, 2006). (b) Exclusive imaging of one population or the other was facilitated by confocal microscopy and by the orientation of muscle being imaged. (c) The morphology of the two types of ICC is distinct: ICC-MY are stellate and form an interconnected network and ICC-DMP are spindle shaped and not connected in a network (Komuro, 2006). After acquisition, video sequences of Ca$^{2+}$ imaging data were imported into custom software (Volumetry G8d, written by G.W. Hennig, University of Vermont). Videos were left unfiltered and Spatio-temporal maps (STMaps) were generated from individual cell ROIs. Quantifiable data was extracted from STMaps using the Fiji/ImageJ, STMapAuto plugin (Leigh et al, 2020) implemented within Fiji/ImageJ (National Institutes of Health, MD, USA, http://rsbweb.nih.gov/ij). Characteristics of Ca$^{2+}$ events were extracted from each STMap and plotted and analyzed using GraphPad Prism version 7.0.0 for Mac,

GraphPad Software, San Diego, California USA, www.graphpad.com.

## Photo-uncaging of IP3 and subsequent imaging

The caged inositol triphosphate, 10 μg, ci-IP3/PM (Cat. No. 6210, Tocris) was prepared to a final concentration of 1.5 μM. Solutions were divided out evenly into 6, 1.5 mL Eppendorf tubes and dissected muscles were cut to 1 cm and placed in each vial for incubation of 1.5-hon ice, then were rinsed with warmed, 37 ± 0.2 °C Krebs solution for 20 min. Micropoint photo stimulation and optogenetics tool (ANDOR Technology, Oxford Instruments) was directed to a region of interest (~5–10 μm in diameter). The laser area was determined and confirmed upon calibration and each laser pulse series was delivered at 10% intensity for a total of 10 s. Each recording was 1.5 min in duration total and recorded at ×60 magnification, see below for confocal spinning disk specifications.

## FRET imaging

The cAMP Encoded Reporter (CAMPER) mice (allele symbol: C57BL/6-Gt(ROSA)26Sor$^{tm1(CAG-ECFP*/Rapgef3/Venus*)Kama/J}$), JAX stock #032205, is a cre recombinase inducible FRET indicator developed to indicate changes in cAMP levels (Muntean et al, 2018). By crossing this mouse with our established Kit-iCre mouse, we were able to successfully induce the EPAC cAMP sensor in KIT$^+$ cells. We utilized a wide-field FRET imaging system that consist of an Eclipse E600FN microscope (Nikon Inc., Melville, NY, USA) equipped with Nikon CFI Fluor lenses (10× 0.3 NA, 20× 0.5 NA, 40× 0.8 NA, 60× 0.8 NA and 100× 1.1 NA; Nikon Instruments, New York, USA). The illumination source of the system is powered by a 150 W Xenon Arc lamp that provides a 6–8 mW with excitation spectra capabilities that range from 250-680 nm excitation (T.I.L.L. Polychrome IV, Grafelfing, Germany). A beam-splitter (DV2 DualView; Photometrics) is connected via a C-mount to one of the microscope's emission ports, which splits the emission light into two (donor and acceptor) channels which can be simultaneously monitored on a single CCD camera chip. Images are detected using a high-speed Andor iXon 897 EMCCD Camera (ANDOR Technology) and allows image acquisition of 33 fps at full frame of 512×512. The cAMP Encoded Reporter (CAMPER) sensor expressing cells were excited with 436 nm light. The emission will split into two channels using the DV2 DualView (505dcxr filter; Photometrics) and was detected at 535 ± 15 nm (Venus) and (mTurquoise) 480 ± 10 nm. FRET changes were monitored using MetaMorph software (MetaMorph Inc, Nashville, TN) and analyzed on dedicated system PC computer.

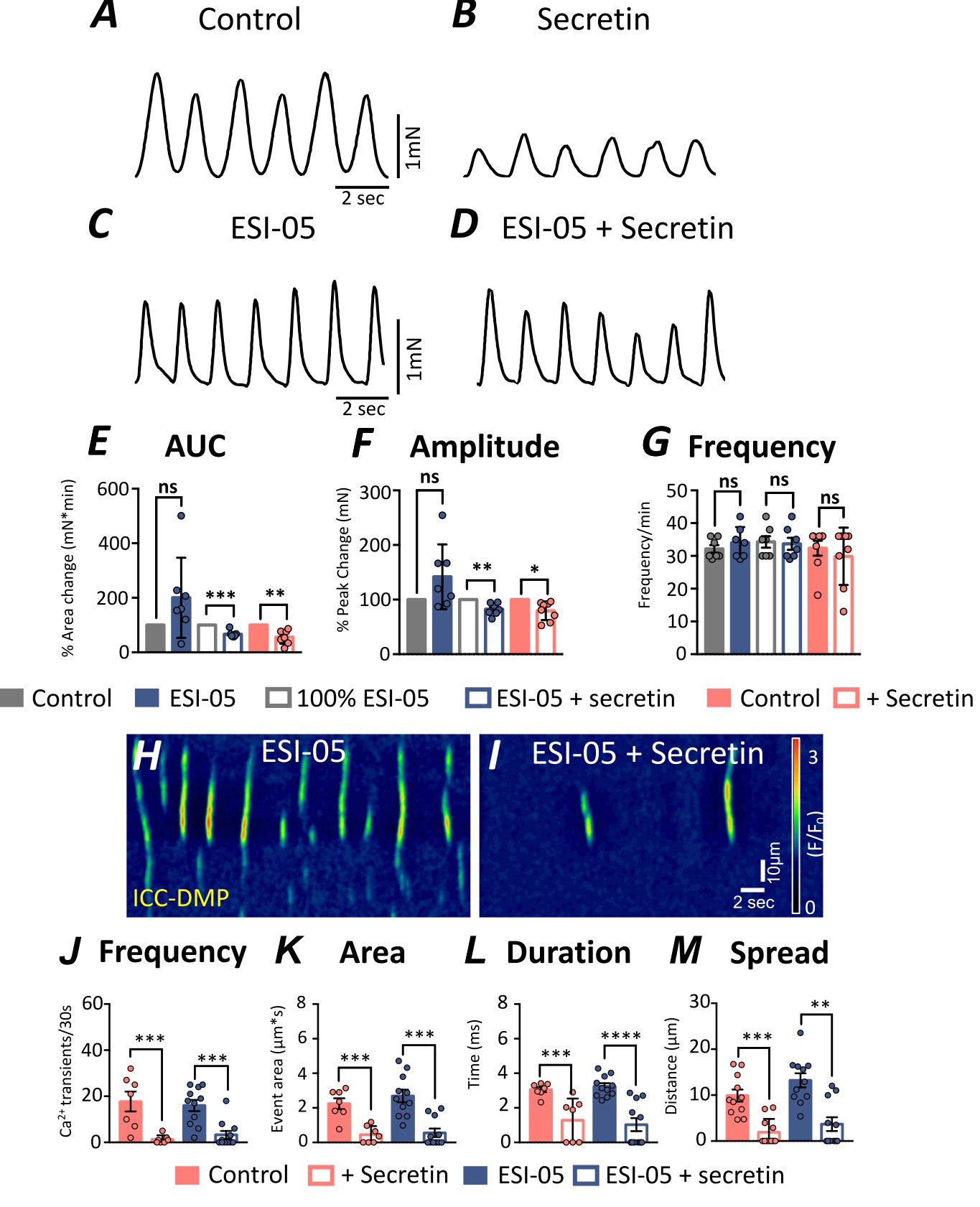

◀ **Figure 13. EPAC antagonist had minimal effects on inhibitory actions of secretin.**

(A, B) Jejunal muscle contractions were inhibited by secretin (100 nM). (C, D) Show the effects of secretin in muscles pretreated with the EPAC antagonist, ESI-05 (10 μM). In the presence of ESI-05 and secretin, the inhibitory effects of secretin on (E) area under the curve (AUC; mN*min) were retained after addition of ESI-05, and no effects were noted on the (F) amplitude of contraction (mN) and (G) frequency (min$^{-1}$) of contractions. (H) Ca$^{2+}$ transients in ICC-DMP plotted as STMaps in the presence of ESI-05 (10 μM) before and (I) after the addition of secretin (100 nM). Inhibition of Ca$^{2+}$ transients in response to secretin persisted in the presence of ESI-05: (J) frequency of Ca$^{2+}$ transients (per 30 s), (K) area of events (mm*s), (L) duration of events (ms), and (M) spatial spread of Ca$^{2+}$ transients (μm) were reduced by secretin before and after addition of ESI-05. The data are plotted as mean ± SEM, and significance was determined using paired *t* test. ***$P = 0.0003$; **$P = 0.0011$ (E); **$P = 0.0058$; *$P = 0.0111$ (F); ***$P = 0.0004$; ***$P = 0.0003$ (J); ***$P = 0.0003$; ***$P = 0.0001$ (K); ***$P = 0.0003$; ****$P < 0.0001$ (L); ***$P = 0.0008$; **$P = 0.0013$, respectively, for all (M). For contraction experiments: secretin alone, strips = 7, $n = 4$ and ESI-05+secretin, strips = 7, $n = 4$. For Ca$^{2+}$ measurement experiments, cells = 11, $n = 5$. Source data are available online for this figure.

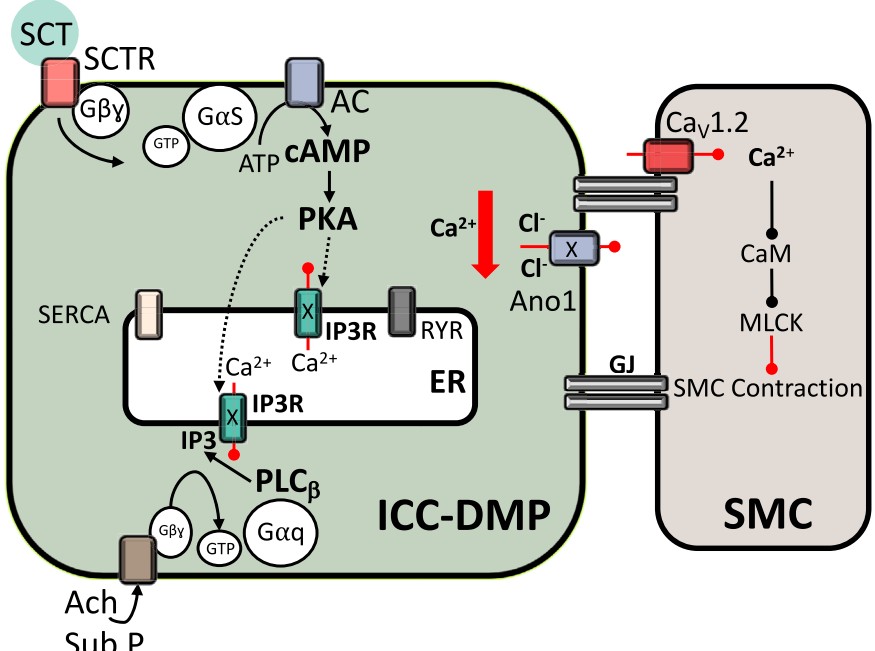

**Figure 14. Secretin mechanisms to inhibit intestinal motility.**

Results in the present study suggest the following mechanism for regulation of small intestinal motility by secretin. Secretin (SCT) binds to the secretin receptor (SCTR) expressed by ICC-DMP in the small intestine and liberates Gαs. Gαs activates adenylate cyclase (AC), leading to production of cyclic AMP (cAMP). cAMP activates protein kinase A (PKA). PKA, by a mechanism yet unknown, inhibits release of Ca$^{2+}$ from inositol 1,4,5-trisphosphate receptors (IP3R). Inhibition of Ca$^{2+}$ release from endoplasmic reticulum (ER) stores, results in reduced availability for intracellular Ca$^{2+}$ and reduced activation of Ca$^{2+}$-activated Cl$^-$ channels (ANO1). Activation of ANO1 produces spontaneous transient inward currents (STICs), that conduct to smooth muscle cells (SMCs) via gap junctions (GJs), producing a depolarizing trend on SMCs and increased open probability of voltage-dependent Ca$^{2+}$ channels (Ca$_V$1.2). Ca$^{2+}$ entry into SMCs binds to calmodulin (CaM), activates myosin light chain kinase (MLCK) and causes contraction. Inhibiting Ca$^{2+}$ release reduces STICs, and the depolarizing influence produced by ICC-DMP, thus reducing Ca$^{2+}$ entry into SMCs and inhibiting contraction. ICC-DMP are transducers of excitatory motor neurotransmitters release from enteric excitatory motor neurons (ACh and neurokinins, such as Substance P (Sub P). The mechanism by which excitatory neural signals are transduced is to increase Ca$^{2+}$ release from IP$_3$ receptors, mediated by increased IP$_3$ production caused by receptor coupling through Gαq and activation of phospholipase Cβ (PLCβ). The effects of secretin were sufficient to largely block the effects of excitatory neurotransmitters on ICC-DMP. These findings suggest the novel concept that ICC-DMP are a focal point for motility of the small intestine, integrating signaling from hormones and motor neurons.

ROIs were drawn across individual cells and fluorescent intensity was recorded. Additionally, background intensities for identical ROI's were recorded and adjusted for the final FRET ratio equation (mTurquoise/Venus), as follows: $(((emTurquoise) \div (eVenus - ((\frac{1}{3}) \times emTurquoise)))$ Where emTurquoise is the background ROI subtracted by the mTurquoise cell ROI; eVenus is the background ROI subtracted by the Venus cell ROI, and bleed-through is adjusted for by ~30%, or 1/3, to correct for most bleed-through effects. Averages for multiple time points were taken for

minimum and maximum FRET values and graphs were plotted using GraphPad Prism version 7.0.0 for Mac, GraphPad Software, San Diego, California USA, www.graphpad.com.

## Statistical analysis

Data were expressed as mean ± SEM in imaging experiments and mean (percentage) ± SEM in contractile recordings, as they were normalized to 100% controls. In imaging experiments, the reported

N values represent mouse number, and corresponding plots include individual points from multiple cells imaged across several tissue preparations. In contractile experiments, the reported $N$ number represents mouse number and individual points represent individual muscle strip preparations recorded from multiple channels. Differences between paired data sets were evaluated by one-way ANOVA or Student's paired $t$ test and differences were considered significant with a $P$ value of $P < 0.05$.

## Data availability

The source data of this paper are collected in the following Dryad database record: https://doi.org/10.5061/dryad.v6wwpzh6q.

The source data of this paper are collected in the following database record: biostudies:S-SCDT-10_1038-S44319-025-00623-1.

## Peer review information

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

## Acknowledgements

Grant Support: This project was supported by R01 DK143049 from the National Institute of Diabetes and digestive and Kidney (NIDDK). We would like to thank the High Spatial and Temporal Resolution Imaging Core at the University of Nevada, Reno School of Medicine (P20GM130459) for their assistance in acquiring images for this study.

## Author contributions

**Allison M Bartlett**: Data curation; Formal analysis; Supervision; Validation; Investigation; Visualization; Methodology; Writing—original draft; Writing—review and editing. **Peter J Blair**: Data curation; Formal analysis; Validation; Investigation; Visualization; Methodology; Writing—original draft. **Kenton M Sanders**: Conceptualization; Resources; Funding acquisition; Investigation; Writing—original draft; Project administration; Writing—review and editing. **Salah A Baker**: Conceptualization; Resources; Software; Supervision; Funding acquisition; Investigation; Methodology; Writing—original draft; Project administration; Writing—review and editing.

Source data underlying figure panels in this paper may have individual authorship assigned. Where available, figure panel/source data authorship is listed in the following database record: biostudies:S-SCDT-10_1038-S44319-025-00623-1.

## Disclosure and competing interests statement

The authors declare no competing interests.

# Expanded View Figures

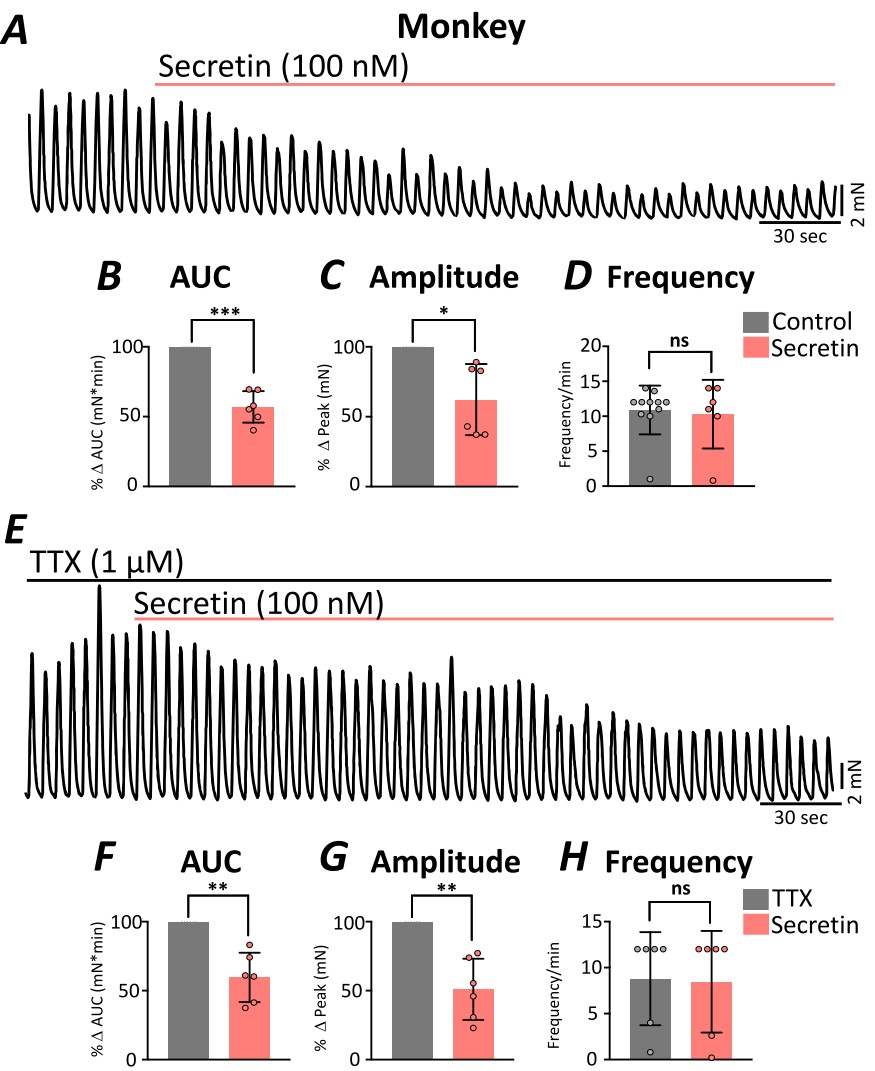

**Figure EV1. Secretin inhibits contractions of small intestinal muscles from Cynomolgus monkeys.**

(**A**) Contractions of monkey jejunal muscles were reduced by secretin (100 nM). (**B**) The area under the curve (AUC; mN*min) and (**C**) amplitude (mN) of contractions were reduced, however, (**D**) the frequency ($min^{-1}$) was unaffected. (**E**) The response to secretin was unaffected by TTX (1μM). (**F**) AUC and (**G**) amplitude of contractions were reduced but (**H**) the frequency of contractions was unaffected. All data were normalized to controls except contraction frequency. The data are plotted as mean ± SEM, and significance was determined using paired *t* test. ***$P = 0.0002$ (**B**); *$P = 0.0150$ (**C**); **$P = 0.0026$ (**F**); **$P = 0.0029$ (**G**); Strips = 7, $n = 3$ for control and strips = 6, $n = 4$ for TTX preparations. Source data are available online for this figure.

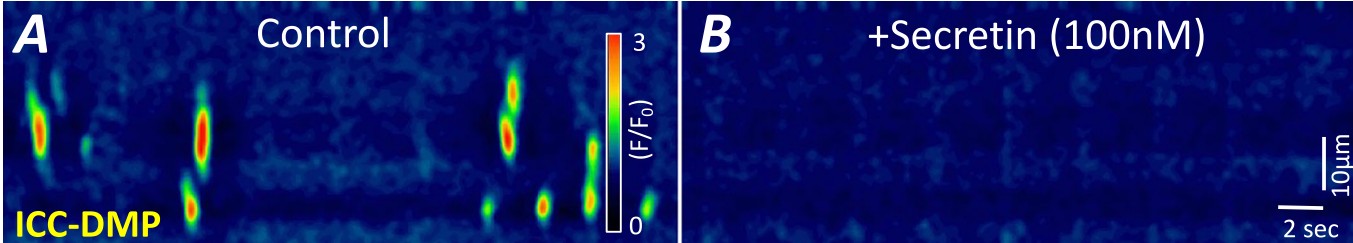

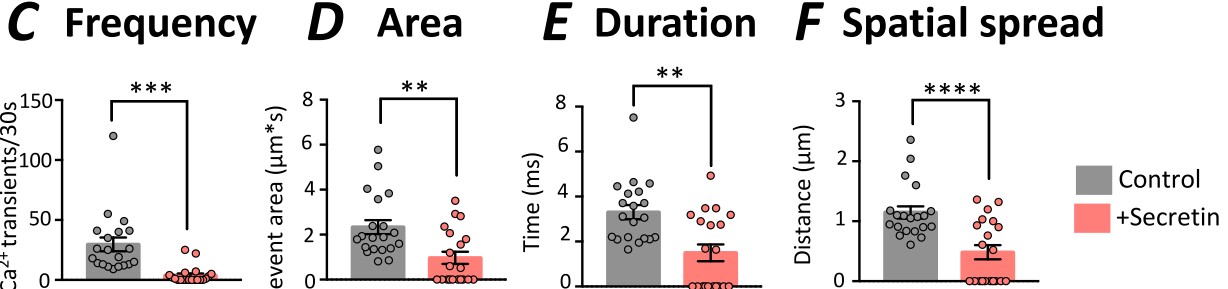

**Figure EV2.  Secretin inhibits Ca²⁺ transients in ICC-DMP.**

Ca²⁺ transients of ICC-DMP were recorded from Kit-iCre-GCaMP6f mouse small intestinal muscles. (**A**) ICC-DMP GCaMP6f signals plotted as STMaps and (**B**) after the addition of secretin (100 nM). Secretin inhibited Ca²⁺ transients: (**C**) frequency of Ca²⁺ transients (per 30 s), (**D**) area of events (μm*s), (**E**) duration of events (ms), and (**F**) spatial spread of Ca²⁺ transients (μm). The data are plotted as mean ± SEM, and significance was determined using paired $t$ test. ***$P = 0.0002$ (**C**); **$P = 0.0015$ (**D**); **$P = 0.0026$ (**E**); ****$P < 0.0001$ (**F**); cells = 20, $n = 10$. Source data are available online for this figure.

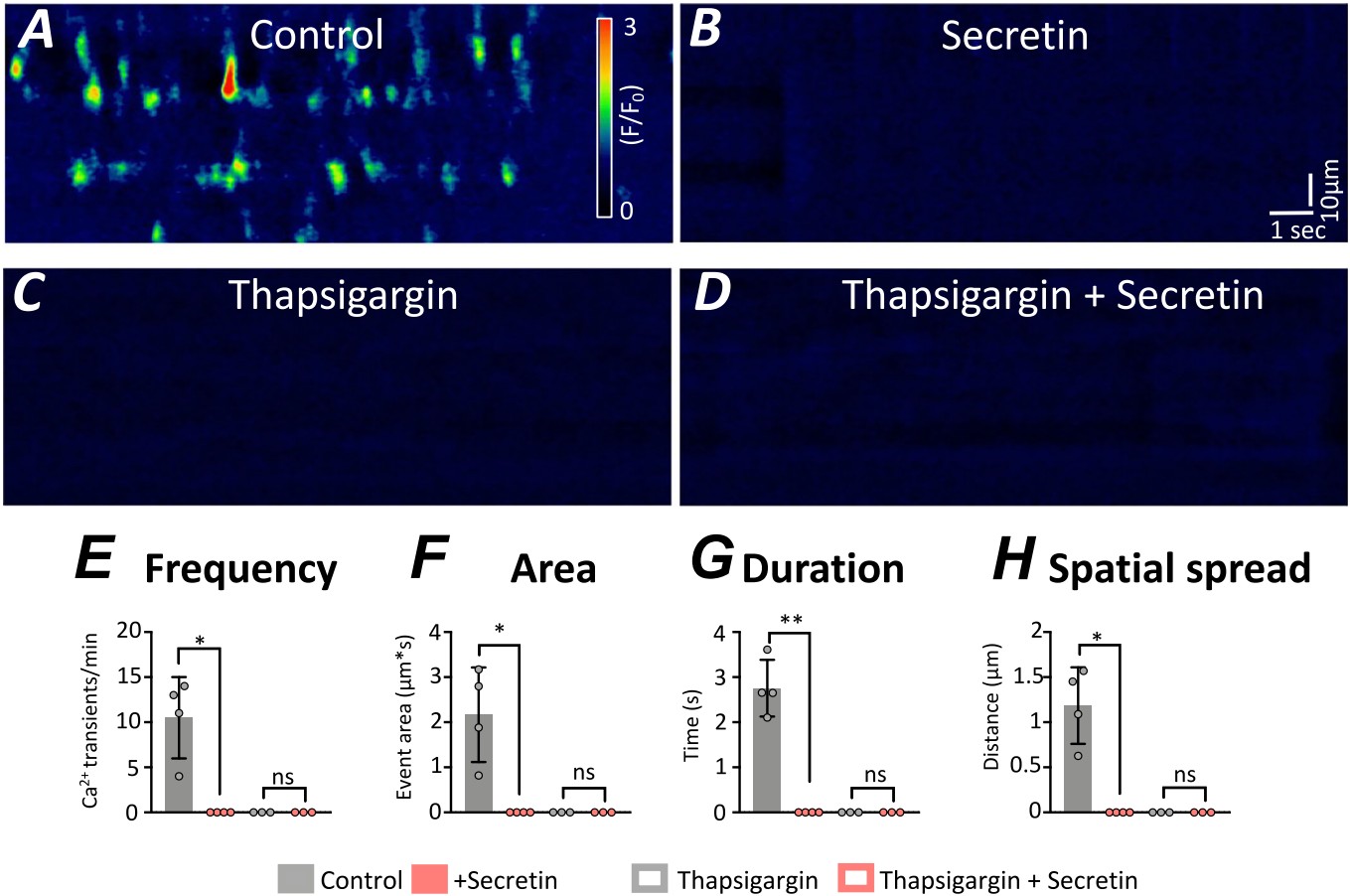

**Figure EV3. Secretin acts on Ca²⁺ release mechanisms in ICC-DMP.**

$Ca^{2+}$ transients of ICC-DMP were recorded from Kit-iCre-GCaMP6f mouse small intestinal muscles. (A) ICC-DMP GCaMP6f signals plotted as STMaps and (B) after the addition of secretin (100 nM). (C) Thapsigargin (10 µM) inhibited $Ca^{2+}$ transients in ICC-DMP, and no further effect was noted after addition of secretin (100 nM) (D). All transients were inhibited: (E) frequency of $Ca^{2+}$ transients (per 30 s), (F) area of events (µm*s), (G) duration of events (ms), and (H) spatial spread of $Ca^{2+}$ transients (µm). The data are plotted as mean ± SEM, and significance was determined using paired $t$ test. *=$P = 0.0187$ (E); *$P = 0.0257$ (F); **$P = 0.0031$ (G); *$P = 0.0113$ (H); for controls, cells = 4, $n = 4$ for thapsigargin, cells = 3, $n = 3$. Source data are available online for this figure.

