## [Peer Review File · EMBO Reports]

Secretin targets interstitial cells of Cajal to regulate intestinal contractions

Allison Bartlett, Peter Blair, Kenton Sanders, and Salah Baker

Corresponding author(s): Salah Baker (sabubaker@med.unr.edu) , Kenton Sanders (ksanders@med.unr.edu)

Review Timeline:

Submission Date:	8th Apr 25
Editorial Decision:	30th May 25
Revision Received:	5th Aug 25
Editorial Decision:	7th Oct 25
Revision Received:	14th Oct 25
Accepted:	21st Oct 25

Editor: Esther Schnapp

Transaction Report:

Dear Prof. Baker,

Thank you for the submission of your manuscript to EMBO reports. We have now received the full set of referee reports that is pasted below.

As you will see, the referees acknowledge that the findings are interesting. However, they also have several suggestions for how the study could be strengthened and improved. I think all suggestions are good and should be addressed. Please let me know in case you disagree, and we can discuss the exact revision requirements further, also in a video chat, if you like.

I would thus like to invite you to revise your manuscript with the understanding that the referee concerns must be fully addressed and their suggestions taken on board. Please address all referee concerns in a complete point-by-point response. Acceptance of the manuscript will depend on a positive outcome of a second round of review. It is EMBO reports policy to allow a single round of major revision only and acceptance or rejection of the manuscript will therefore depend on the completeness of your responses included in the next, final version of the manuscript.

We realize that it is difficult to revise to a specific deadline. In the interest of protecting the conceptual advance provided by the work, we recommend a revision within 3 months (30th Aug 2025). Please discuss the revision progress ahead of this time with the editor if you require more time to complete the revisions.

- 1) A data availability section providing access to data deposited in public databases is missing. If you have not deposited any data, please add a sentence to the data availability section that explains that.
- 2) Your manuscript contains statistics and error bars based on $n=2$. Please use scatter blots in these cases. No statistics should be calculated if $n=2$.

5) a complete author checklist, which you can download from our author guidelines <https://www.embopress.org/page/journal/14693178/authorguide>. Please insert information in the checklist that is also reflected in the manuscript. The completed author checklist will also be part of the RPF.

6) Please note that all corresponding authors are required to supply an ORCID ID for their name upon submission of a revised manuscript (<https://orcid.org/>). Please find instructions on how to link your ORCID ID to your account in our manuscript tracking system in our Author guidelines <https://www.embopress.org/page/journal/14693178/authorguide#authorshipguidelines>

7) Before submitting your revision, primary datasets produced in this study need to be deposited in an appropriate public

database (see <https://www.embopress.org/page/journal/14693178/authorguide#datadeposition>). Please remember to provide a reviewer password if the datasets are not yet public. The accession numbers and database should be listed in a formal "Data Availability" section placed after Materials & Method (see also <https://www.embopress.org/page/journal/14693178/authorguide#datadeposition>). Please note that the Data Availability Section is restricted to new primary data that are part of this study. * Note - All links should resolve to a page where the data can be accessed. *

- the name of the statistical test used to generate error bars and P values,
- the number (n) of independent experiments (please specify technical or biological replicates) underlying each data point,
- the nature of the bars and error bars (s.d., s.e.m.),
- If the data are obtained from n Program fragment delivered error `Can't locate object method "less" via package "than" (perhaps you forgot to load "than"?) at //ejpvfs23/sites23b/embor_www/letters/embor_decision_revise_and_review.txt line 56.' 2, use scatter blots showing the individual data points.

12) All Materials and Methods need to be described in the main text using our 'Structured Methods' format, which is required for all research articles. According to this format, the Methods section includes a Reagents and Tools Table (listing key reagents, experimental models, software and relevant equipment and including their sources and relevant identifiers) followed by a Methods and Protocols section describing the methods using a step-by-step protocol format. The aim is to facilitate adoption of the methodologies across labs. More information on how to adhere to this format as well as a downloadable template (.docx) for the Reagents and Tools Table can be found in our author guidelines: <https://www.embopress.org/page/journal/14693178/authorguide#structuredmethods>.

An example of a Method paper with Structured Methods can be found here: <https://www.embopress.org/doi/full/10.1038/s44320-024-00037-6#sec-4>

You are able to opt out of this by letting the editorial office know (emboreports@embo.org). If you do opt out, the Review Process File link will point to the following statement: "No Review Process File is available with this article, as the authors have

chosen not to make the review process public in this case."

I look forward to seeing a revised form of your manuscript when it is ready.

Referee #1:

The paper of Bartlett and colleagues is a well-designed and thoroughly conducted study demonstrating that secretin acts on the ICC in the deep muscular plexus (ICC-DMP) to reduce motility. The mechanism of action is $G\alpha_s$ -coupled cAMP production and PKA activation, leading to inhibition of IP3 receptors. The findings of the study are novel and interesting, and the authors combined classical, established methods to assess motility with cutting edge technologies. The main finding of the study adds significant knowledge to basic gastrointestinal physiology, the results are convincingly presented and extensively illustrated.

Still, I have some questions and comments.

Main questions/comments:

1. When inhibiting major enteric inhibitory neurotransmission during electrical stimulation in jejunal preparations, the authors inhibited nitric oxide production and a purine receptor, but did not discuss the possible role of VIP. Can VIP play a role?
2. The authors use two different settings for electrical stimulation, 3 Hz, 15s period and 10 Hz for 10 s. What is the rationale behind this?
3. Page 10 Line 2-13. The experiments with Carbachol need more explanation. Without the whole context, these results could also suggest that besides the ICC-DMP, secretin may also act directly on the smooth muscle cells (SMC) to inhibit / modulate the cholinergic excitatory input. The argument to rule this out is that no SCTR has been found on the SMC? I find the explanation of the results: "These data show that secretin inhibits contractile responses to whole-muscle excitatory stimulation via a muscarinic mechanism." too short. It may mislead the readers suggesting it blocks muscarinic receptors for example. Could the authors please extend the explanation of these experiments?
4. I find the description of the results with ESI-05 somehow confusing. When contemplating the results, it appears that the effect of secretin on the amplitude was abolished, whereas on the AUC the effect is reduced, but still present. The results text describes these findings, yet in a way that is difficult to follow. Can the authors reformulate the text to be more fluent and comment on the significance of this finding?

Minor comments and suggestions:

- Page 10 Line 2: Secretin reduced the force of contractions jejunal muscles pre-stimulated with carbachol" should be "in" or "of" jejunal muscles
- Page 10 Line 10: "The" should be "the"
- Page 10 line 14: the word "stimulation" should be deleted
- Page 11 line 12: "nominal-Ca²⁺ Krebs solution": does that mean a "nominal zero Ca²⁺ Krebs solution"? To the best of my knowledge, nominal concentration means "the target concentration, which may not be the exact same as the actual concentration". But used in this form, it is not obvious to me what the exact meaning is.
- Page 15 line 16 "reduced There" the dot is missing
- Page 27 Line 15 "were" is missing
- Figure legends:
 - o Figure 1. and Suppl. Figure 1 "secretin inhibit" should be "inhibits"
 - o Figure 5: "tin" should be "in", "continues" should be "continuous"
 - o Suppl. Figure 2: "(D) with the additional application of secretin (100nM)." Is here the verb missing?
- Figure 15: the red "sticks" are mostly behind the green rectangles, but the one at the lower side of the ER is over the rectangle. Is that intentional?
- Supplementary table 1: can the authors double check the p value of frequency in the first row? It may be correct, but without seeing the original values it seems strange that with normal distribution, not extremely high n-numbers (only 8) and such similar mean + SEM, and dot distribution in the figure, the p value is so close to 0.05.

Referee #2:

In this manuscript Bartlett and colleagues describe their experiments investigating the mechanisms whereby secretin, a gut hormone important for digestive function, slows down gastrointestinal motility. They combine a variety of physiology (smooth muscle contractility) techniques, advanced fluorescence imaging methods, chemogenetics and pharmacological interventions on mouse and monkey jejunum. The paper is generally well written and nicely illustrated. This study uncovers that secretin acts on a specific subtype of interstitial cells of Cajal (those of the deep muscular plexus), inhibiting IP3 receptor-mediated intracellular calcium transients. This is an exciting and highly novel finding but I do have some remarks and questions that require clarification.

A concentration 100 nM of secretin is used. How was this concentration chosen and how does it compare to levels found in the blood (postprandially)?

The jejunum was used in this study, which is a logical choice. However, do the authors have info on more distal intestinal regions, at both the functional and expression level? Is Sctr expressed exclusively in ICC-DMP in other parts of the small intestine?

In a single cell transcriptomic paper, it was reported that SCTR is also expressed on human enteric neurons (10.1016/j.cell.2020.08.003). It would be good to mention this.

Focusing on the mouse ortholog of the secretin receptor, please use Sctr.

All contractility traces are shown until the end of secretin stimulation. What happens upon washout? Also for CNO, 8-bromo-cAMP,...

How do the contractility parameters compare between stimulation using secretin alone with stimulation with secretin in combination with TTX (Figure 1: B-D vs F-H).

The rationale of the experiment with 'nominal' extracellular calcium is not clear to me. If nominal calcium does not affect intracellular transients in the first place, than why test the effect of secretin on top of that? In Figure 8 it would be good to show on the STMs when secretin was puffed.

Why was TTX added for the calcium imaging experiments using 8-bromo-cAMP?

In the results section it is mentioned that FRET efficiency in response to forskolin and secretin increase to 115% and 108% respectively. These values don't correspond with what is shown in Figure 12.

Referee #3:

The current study reveals a novel, non-neural mechanism through which the gastrointestinal hormone secretin regulates small intestinal smooth muscle contraction. Specifically, the authors identify that secretin receptors (SCTR) are highly expressed on interstitial cells of Cajal located in the deep muscular plexus (ICC-DMP). Functionally, secretin was found to inhibit small intestinal smooth muscle contractions and suppress excitatory enteric neurotransmission. Mechanistically, this inhibition appears to occur via a G α s-coupled signaling cascade involving cAMP production, PKA activation, and inhibition of IP3 receptor mediated calcium release in ICC-DMP. These findings support a previously unrecognized role for ICC-DMP.

The study has several positive aspects. It introduces a non-neural role for secretin in regulating GI function. Through rigorous imaging analyses, the authors delineate a clear signaling cascade involving cAMP and PKA downstream of SCTR in ICC-DMP. The findings contribute valuable insight into calcium signaling dynamics in ICC-DMP and the broader understanding of intestinal motility regulation. Furthermore, the use of multiple experimental models strengthens the findings.

Despite its strengths, the study has several important weak points. The mechanism linking PKA activation to suppression of IP3 mediated calcium release remains speculative. Experimental evidence confirming this step is lacking. Functional outcomes were demonstrated only in ex vivo or in vitro preparations. The absence of in vivo validation limits the physiological relevance of the findings. Expression of SCTR in other gastrointestinal cell types-such as smooth muscle cells-was not evaluated, leaving open the possibility that secretin may act through additional pathways to influence motility.

Overall, this study provides a compelling new framework for understanding how hormonal signals, particularly secretin, integrate with ICC function in the gut to modulate motility. The identification of ICC-DMP as direct targets of secretin opens new avenues for research into non-neural regulation of GI function. Addressing the following points would further improve the significance of this study.

Major comments:

1. Since in vivo experiments were not performed in this study, the title should be revised to reflect the focus on ex vivo or cellular mechanisms rather than intestinal motility.
2. It remains unclear whether other cell types within the GI muscularis express SCTR. The observed inhibition of carbachol-induced contraction suggests the possibility of a direct effect on smooth muscle cells, which should be discussed as a limitation.
3. The study proposes a novel mechanism for secretin in ICC-DMP - activation of G α s by secretin - increased cAMP production - PKA activation - inhibition of IP3R - decreased calcium release by ICC-DMP. However, Figure 7 implies a possible direct effect of secretin on IP3Rs. Clarification is needed: does secretin act solely through these signaling pathway, or could it also antagonize IP3Rs directly?

Minor comments:

1. Supplementary figures should be reorganized and numbered in accordance with their order of appearance in the results

section for clarity and consistency.

2. The data on monkey jejunum do not appear to contribute additional mechanistic insights and could be omitted or moved to supplementary materials unless further justified.

3. The lack of effect of AT7867 alone in Figure 13 is surprising. Given that PKA activation reduces both calcium signaling and contraction (Figure 8-11), PKA inhibition would be expected to produce the opposite effect. This discrepancy needs further discussion or investigation.

Referee #1:

The paper of Bartlett and colleagues is a well-designed and thoroughly conducted study demonstrating that secretin acts on the ICC in the deep muscular plexus (ICC-DMP) to reduce motility. The mechanism of action is G α s-coupled cAMP production and PKA activation, leading to inhibition of IP $_3$ receptors. The findings of the study are novel and interesting, and the authors combined classical, established methods to assess motility with cutting edge technologies. The main finding of the study adds significant knowledge to basic gastrointestinal physiology, the results are convincingly presented and extensively illustrated. Still, I have some questions and comments.

Thank you for your positive comments and suggestions to improve the manuscript. We have addressed each point in the manner described below.

Main questions/comments:

1. When inhibiting major enteric inhibitory neurotransmission during electrical stimulation in jejunal preparations, the authors inhibited nitric oxide production and a purine receptor, but did not discuss the possible role of VIP. Can VIP play a role?

VIP and PACAP can contribute as inhibitory enteric neurotransmitters in the GI tract. But these contribute a minor component of enteric inhibitory neurotransmission in small intestine at the frequencies of stimulation used in our experiments, as inhibitory neurotransmission is completely blocked by inhibition of NO synthesis and blockade of P $_2$ Y $_1$ receptors (Baker *et al.*, 2018).

The purpose of our study was to investigate post-junctional effects of secretin based on the observation of robust expression of secretin receptors by ICC-DMP. A recent report has suggested expression of secretin by neurons of the enteric nervous system (Drokhlyansky *et al.*, 2020), although the function of these receptors and possible neural effects in smooth muscle layers is unknown. We used TTX, a commonly used neurotoxin, to eliminate contamination from effects of secretin possibly mediated through enteric neurons and conveyed to SMCs via nerve action potentials and neurotransmission. An inhibitor of NO synthesis and an antagonist of P $_2$ Y $_1$ receptors were included in one of these experiments as additional insurance that the inhibitory effects of secretin were not mediated by enteric inhibitory neurotransmission.

2. The authors use two different settings for electrical stimulation, 3 Hz, 15s period and 10 Hz for 10 s. What is the rationale behind this?

We wanted to determine whether excitatory nerve stimulation could overcome the effects of secretin. Therefore, we first tested whether nerve stimulation at a moderate stimulus intensity (3 Hz) could overcome the effects of secretin, and this did not occur. We explained a weakness in this experiment was that EFS of GI muscles causes release of both excitatory and inhibitory neurotransmitters. Thus, release of NO and

purines might complement the inhibitory effects of secretin. So, we repeated the same experiment with these dominant inhibitory neurotransmitters blocked. We also increased the intensity of EFS (10 Hz) to more intensely stimulate remaining excitatory neurotransmission. We found that the inhibitory effects of secretin were sustained even with strong excitatory nerve stimulation. The important and novel point made is that secretin, which is secreted when chyme enters the small intestine, can reduce the post-junctional influence of excitatory neurotransmission. Reducing excitatory effects may provide an important 'brake' on small intestinal transit to increase time for neutralization of acid and digestion and absorption of nutrients. We have added more clarification of our rationale for using the 2 frequencies of stimulation in the Results section.

3. Page 10 Line 2-13. The experiments with Carbachol need more explanation. Without the whole context, these results could also suggest that besides the ICC-DMP, secretin may also act directly on the smooth muscle cells (SMC) to inhibit / modulate the cholinergic excitatory input. The argument to rule this out is that no SCTR has been found on the SMC? I find the explanation of the results: "These data show that secretin inhibits contractile responses to whole-muscle excitatory stimulation via a muscarinic mechanism." too short. It may mislead the readers suggesting it blocks muscarinic receptors for example. Could the authors please extend the explanation of these experiments?

RNAseq performed on sorted SMCs and ICC from the small intestine showed very low expression of secretin receptor in SMC, as compared to ICC: e.g 0.8 FPKM in SMC vs. 102.2 FPKM in ICC (Lee *et al.*, 2015; Lee *et al.*, 2017). To further test these findings, we performed RNAscope in the present study and did not detect expression of secretin receptors in SMC (see Figure 2). Thus, direct effects of secretin on SMC are unlikely. As per the reviewer requests, we clarified this point in the Introduction and Results.

4. I find the description of the results with ESI-05 somehow confusing. When contemplating the results, it appears that the effect of secretin on the amplitude was abolished, whereas on the AUC the effect is reduced, but still present. The results text describes these findings, yet in a way that is difficult to follow. Can the authors reformulate the text to be more fluent and comment on the significance of this finding?

Per the reviewer's suggestion, we have clarified these findings. The inhibitory effects of secretin in the presence of ESI-05 were generally observed. In some instances, ESI-05 caused frequent changes in contractile amplitude, as shown in revised Figure 13D. We reanalyzed our data to incorporate this observation. Overall, both the AUC and the amplitude of contraction were reduced in the presence of ESI-05. These results suggest that EPAC may play a minor role in mediating the secretin response, especially when compared to PKA antagonism.

This clarification has been added to the Results section.

Minor comments and suggestions:

- Page 10 Line 2: Secretin reduced the force of contractions jejunal muscles pre-stimulated with carbachol" should be "in" or "of" jejunal muscles

Corrected.

- Page 10 Line 10: "The" should be "the"

Corrected.

- Page 10 line 14: the word "stimulation" should be deleted

Extra word deleted.

- Page 11 line 12: "nominal-Ca²⁺ Krebs solution": does that mean a "nominal zero Ca²⁺ Krebs solution"? To the best of my knowledge, nominal concentration means "the target concentration, which may not be the exact same as the actual concentration". But used in this form, it is not obvious to me what the exact meaning is.

Definition of what we meant by 'nominally Ca²⁺ free' was added to the Results and Methods.

- Page 15 line 16 "reduced There" the dot is missing

Corrected.

- Page 27 Line 15 "were" is missing

Corrected.

- Figure legends:

o Figure 1. and Suppl. Figure 1 "secretin inhibit" should be "inhibits"

Both corrected.

o Figure 5: "tin" should be "in", "continues" should be "continuous"

Both corrected.

o Suppl. Figure 2: "(D) with the additional application of secretin (100nM)." Is here the verb missing?

Statement was corrected.

- Figure 15: the red "sticks" are mostly behind the green rectangles, but the one at the lower side of the ER is over the rectangle. Is that intentional?

Sticks were fixed.

- Supplementary table 1: can the authors double check the p value of frequency in the first row? It may be correct, but without seeing the original values it seems strange that with normal distribution, not extremely high n-numbers (only 8) and such similar mean + SEM, and dot distribution in the figure, the p value is so close to 0.05.

The P value is correct. We used paired t-test to analyze the data (Appendix Table S1).

Referee #2:

In this manuscript Bartlett and colleagues describe their experiments investigating the mechanisms whereby secretin, a gut hormone important for digestive function, slows down gastrointestinal motility. They combine a variety of physiology (smooth muscle contractility) techniques, advanced fluorescence imaging methods, chemogenetics and pharmacological interventions on mouse and monkey jejunum. The paper is generally well written and nicely illustrated. This study uncovers that secretin acts on a specific subtype of interstitial cells of Cajal (those of the deep muscular plexus), inhibiting IP3 receptor-mediated intracellular calcium transients. This is an exciting and highly novel finding but I do have some remarks and questions that require clarification.

Thank you for your supportive comments and many excellent suggestions for revision of our manuscript. Changes made in the revision are detailed in a point-by-point manner below.

A concentration of 100 nM secretin was used because its effect was consistently reproducible in contraction experiments compared to lower concentrations. Please see new Appendix Figure S1. Secretin at 5 nM had no significant effects, and only small effects were observed at 10 nM. Several studies have also reported that a concentration of 100 nM is effective in mice (see references below):

1. Jessica Y. S. Chu et al., 2009, *PNAS*: <https://doi.org/10.1073/pnas.0903695106> (Chu et al., 2009)
2. Veronika Csillag et al., 2019, *Frontiers in Cellular Neuroscience*: <https://www.frontiersin.org/articles/10.3389/fncel.2019.00371/full> (Csillag et al., 2019)
3. Shannon Glaser et al., 2010, *Hepatology*: https://journals.lww.com/hep/abstract/2010/07000/knockout_of_secretin_receptor_reduces_large.22.aspx (Glaser et al., 2010)

While a concentration closer to endogenous secretin levels would be ideal, the potential for peptide degradation in our experimental conditions, along with peptidase activity that may diminish secretin's bioavailability, supports the use of a higher concentration. Therefore, we utilized 100 nM secretin to ensure sufficient and consistent activation of target receptors in gastrointestinal muscle preparations, providing robust and reproducible physiological effects and similar to previously published papers listed above.

We added this information to the results section.

- The jejunum was used in this study, which is a logical choice. However, do the authors have info on more distal intestinal regions, at both the functional and expression level? Is Sctr expressed exclusively in ICC-DMP in other parts of the small intestine?

We have not evaluated regional differences in secretin receptor expression across the small intestine.

In a single cell transcriptomic paper, it was reported that SCTR is also expressed on human enteric neurons (10.1016/j.cell.2020.08.003). It would be good to mention this.

As the reviewer suggest, we added a comment about this in the Introduction, and reinforced this in the Discussion.

- Focusing on the mouse ortholog of the secretin receptor, please use Sctr.

We have edited the paper using *Sctr* to express gene transcripts and SCTR when discussing the protein. We think this is consistent throughout the paper now.

All contractility traces are shown until the end of secretin stimulation. What happens upon washout? Also for CNO, 8-bromo-cAMP,...

Secretin washed out quickly, typically within 10–15 minutes after removal. We have added a new Appendix Figure S2 to illustrate the effects of secretin and its washout on intestinal contractions.

Additionally, both CNO and 8-bromo-cAMP washed out without any issues. We added this information to the results section.

How do the contractility parameters compare between stimulation using secretin alone with stimulation with secretin in combination with TTX (Figure 1: B-D vs F-H).

There is no significant difference between responses to secretin before or after TTX was added. We have updated Figure 1 to include a comparison of the effects of secretin with and without TTX, and we have added a statement in the results section to reflect this. This comparison is now shown in the revised Figure 1.

-The rationale of the experiment with 'nominal' extracellular calcium is not clear to me. If nominal calcium does not affect intracellular transients in the first place, than why test the effect of secretin on top of that? In Figure 8 it would be good to show on the STMs when secretin was puffed.

-Store-operated calcium entry via ORAI channels has been shown to be important for Ca^{2+} signaling in ICC-DMP (Zheng *et al.*, 2018). Therefore, we conducted these experiments in nominally Ca^{2+} free Krebs solution (now defined in Methods) to determine whether secretin may influence Ca^{2+} entry through this pathway. Our data suggest that ORAI-mediated calcium entry does not contribute to the secretin response.
-When secretin was applied via puff, an artifact in the imaging was observed. Thus, we eliminated the artifact and show only the image data captured immediately after the puff application.

-Why was TTX added for the calcium imaging experiments using 8-bromo-cAMP?
We applied TTX to block nerve action potentials and thereby minimize any potential contamination from effects of the cAMP analogue on myenteric motor neurons. This approach helped to isolate direct effects 8-bromo-cAMP on post-junctional cells of the SIP syncytium. Imaging showed direct effects on ICC-DMP.

- In the results section it is mentioned that FRET efficiency in response to forskolin and secretin increase to 115% and 108% respectively. These values don't correspond with what is shown in Figure 12.

We have updated the figure summary graphs with normalized FRET results to 100% of controls and we updated the percentages in Appendix Table 10.

Referee #3:

The current study reveals a novel, non-neural mechanism through which the gastrointestinal hormone secretin regulates small intestinal smooth muscle contraction. Specifically, the authors identify that secretin receptors (SCTR) are highly expressed on interstitial cells of Cajal located in the deep muscular plexus (ICC-DMP). Functionally, secretin was found to inhibit small intestinal smooth muscle contractions and suppress excitatory enteric neurotransmission. Mechanistically, this inhibition appears to occur via a G α s-coupled signaling cascade involving cAMP production, PKA activation, and inhibition of IP3 receptor mediated calcium release in ICC-DMP. These findings support a previously unrecognized role for ICC-DMP.

The study has several positive aspects. It introduces a non-neural role for secretin in regulating GI function. Through rigorous imaging analyses, the authors delineate a clear signaling cascade involving cAMP and PKA downstream of SCTR in ICC-DMP. The findings contribute valuable insight into calcium signaling dynamics in ICC-DMP and the broader understanding of intestinal motility regulation. Furthermore, the use of multiple experimental models strengthens the findings.

Despite its strengths, the study has several important weak points. The mechanism linking PKA activation to suppression of IP3 mediated calcium release remains speculative. Experimental evidence confirming this step is lacking. Functional outcomes were demonstrated only in ex vivo or in vitro preparations. The absence of in vivo validation limits the physiological relevance of the findings. Expression of SCTR in other gastrointestinal cell types-such as smooth muscle cells-was not evaluated, leaving open the possibility that secretin may act through additional pathways to influence motility.

Overall, this study provides a compelling new framework for understanding how hormonal signals, particularly secretin, integrate with ICC function in the gut to modulate motility. The identification of ICC-DMP as direct targets of secretin opens new avenues for research into non-neural regulation of GI function. Addressing the following points would further improve the significance of this study.

Thank you for your many supportive comments and suggestions for revisions. Changes made in the revised manuscript are detailed below.

Major comments:

1. Since in vivo experiments were not performed in this study, the title should be revised to reflect the focus on ex vivo or cellular mechanisms rather than intestinal motility. Per reviewer suggestion, we changed the title to “Secretin targets interstitial cells of Cajal to regulate intestinal contractions”
2. It remains unclear whether other cell types within the GI muscularis express SCTR.

The observed inhibition of carbachol-induced contraction suggests the possibility of a direct effect on smooth muscle cells, which should be discussed as a limitation. RNAseq performed on sorted SMCs and ICC from the small intestine showed extremely low expression of the secretin receptor in SMC compared to ICC: 0.8 FPKM in SMC vs. 102.2 FPKM in ICC. In confirmation of these findings we did not detect expression of secretin receptors in SMC using RNAscope (see Figure 2). Thus, direct effects of secretin on SMC are unlikely. We clarified this point in the Introduction and Results. Experiments in Figure 4 showed that excitatory nerve stimulation could not overcome potent inhibitory effects of secretin. Experiments testing exogenous carbachol (CCh) would surely activate muscarinic receptors expressed by both ICC and SMCs. We wanted to see if effects of secretin are still present with additional excitatory effects of CCh. Our data show that the potent inhibitory effects of secretin on ICC are sustained even during stimulation with CCh.

3. The study proposes a novel mechanism for secretin in ICC-DMP - activation of $G_{\alpha s}$ by secretin - increased cAMP production - PKA activation - inhibition of IP3R - decreased calcium release by ICC-DMP. However, Figure 7 implies a possible direct effect of secretin on IP3Rs. Clarification is needed: does secretin act solely through these signaling pathway, or could it also antagonize IP3Rs directly?

We are not sure what the reviewer is suggesting as a direct effect of secretin on IP3 receptors. Secretin is a peptide that is unlikely to enter cells directly. Its effects most likely are mediated through binding to secretin receptors in the plasma membrane and then initiating effects via 2nd messengers. In the case of secreting this most likely occurs through liberation of $G_{\alpha s}$ and generation of cAMP. Our data suggest that activation of PKA is also responsible, however at this point we do not understand the linkage between PKA and IP3 receptors. This part of the mechanism will require much more study, as there may be an unknown intermediate protein phosphorylated by PKA that causes inhibition of IP3 receptors. We have changed Figure 14 and statements in the Figure legend to clarify these points.

Minor comments:

1. Supplementary figures should be reorganized and numbered in accordance with their order of appearance in the results section for clarity and consistency.

All supplementary figures and materials were reorganized as requested

2. The data on monkey jejunum do not appear to contribute additional mechanistic insights and could be omitted or moved to supplementary materials unless further justified.

Per reviewer suggestion, we moved this figure to supplemental materials as Figure EV1.

3. The lack of effect of AT7867 alone in Figure 13 is surprising. Given that PKA activation reduces both calcium signaling and contraction (Figure 8-11), PKA inhibition would be expected to produce the opposite effect. This discrepancy needs further discussion or investigation.

The PKA inhibitor AT7867 alone occasionally showed an increase in AUC and contractile amplitude; however, these changes did not reach statistical significance (revised Figure 12E&F). We agree with the reviewer observation, we expected to see an increase in contractile activity, but this was not the case as the data did not reach significance. PKA-mediated mechanisms are complex and can vary between cell types, which may have influenced the observed data. Additionally, other PKA-dependent pathways could have affected the results. For instance, PKA-mediated phosphorylation of phospholamban (PLN) enhances SERCA activity, promoting calcium reuptake and contributing to the regulation and dampen of contractile dynamics.

Literature cited

- Baker SA, Drumm BT, Cobine CA, Keef KD & Sanders KM. (2018). Inhibitory Neural Regulation of the Ca (2+) Transients in Intramuscular Interstitial Cells of Cajal in the Small Intestine. *Front Physiol* **9**, 328.
- Chu JY, Lee LT, Lai CH, Vaudry H, Chan YS, Yung WH & Chow BK. (2009). Secretin as a neurohypophysial factor regulating body water homeostasis. *Proc Natl Acad Sci U S A* **106**, 15961-15966.
- Csillag V, Vastagh C, Liposits Z & Farkas I. (2019). Secretin Regulates Excitatory GABAergic Neurotransmission to GnRH Neurons via Retrograde NO Signaling Pathway in Mice. *Front Cell Neurosci* **13**, 371.
- Drokhlyansky E, Smillie CS, Van Wittenberghe N, Ericsson M, Griffin GK, Eraslan G, Dionne D, Cuoco MS, Goder-Reiser MN, Sharova T, Kuksenko O, Aguirre AJ, Boland GM, Graham D, Rozenblatt-Rosen O, Xavier RJ & Regev A. (2020). The Human and Mouse Enteric Nervous System at Single-Cell Resolution. *Cell* **182**, 1606-1622.e1623.
- Glaser S, Lam IP, Franchitto A, Gaudio E, Onori P, Chow BK, Wise C, Kopriva S, Venter J, White M, Ueno Y, Dostal D, Carpino G, Mancinelli R, Butler W, Chiasson V, DeMorrow S, Francis H & Alpini G. (2010). Knockout of secretin receptor reduces large cholangiocyte hyperplasia in mice with extrahepatic cholestasis induced by bile duct ligation. *Hepatology* **52**, 204-214.
- Lee MY, Ha SE, Park C, Park PJ, Fuchs R, Wei L, Jorgensen BG, Redelman D, Ward SM, Sanders KM & Ro S. (2017). Transcriptome of interstitial cells of Cajal reveals unique and selective gene signatures. *PLoS One* **12**, e0176031.
- Lee MY, Park C, Berent RM, Park PJ, Fuchs R, Syn H, Chin A, Townsend J, Benson CC, Redelman D, Shen TW, Park JK, Miano JM, Sanders KM & Ro S. (2015). Smooth Muscle Cell Genome Browser: Enabling the Identification of Novel Serum Response Factor Target Genes. *PLoS One* **10**, e0133751.
- Zheng H, Drumm BT, Earley S, Sung TS, Koh SD & Sanders KM. (2018). SOCE mediated by STIM and Orai is essential for pacemaker activity in the interstitial cells of Cajal in the gastrointestinal tract. *Sci Signal* **11**.

Dear Prof. Baker,

Thank you for the submission of your revised manuscript. We have now received the enclosed reports from the referees and I am happy to say that all support its publication now. Only referee 1's comment will need to be addressed, as well as some editorial requests before we can proceed with the official acceptance of your manuscript.

- Please send us a completed author checklist, which you can download from our author guidelines <<https://www.embopress.org/page/journal/14693178/authorguide>>, with your final ms. The current checklist is missing all answers from you. The completed author checklist will also be part of your transparent peer-review file.
- Please reduce the number of keywords to 5.
- Please list the funding info under the Acknowledgements.
- The author credits need to be removed from the ms file. All credits need to be entered during online ms submission.
- Please remove the movie legend from the manuscript text and add it to a text or .doc file, then ZIP this with the movie file and upload the zip folder. The correct nomenclature is "Movie EV1".
- Please add the appendix figure legends to the Appendix file directly following their respective Appendix figure.
- There is a folder labelled Fig 8 I-J in the source data ZIP folder for Fig 7, this might be mislabelled? The source data need to be uploaded as one folder per main figure. The source data for EV and Appendix figures can be combined into one folder. Please add the two folders with the secretin data to the corresponding source data folders for the figures.
- FIGURE CALLOUTS are missing for Fig 2A, Fig 5D, Fig 8I,J,K, Fig EV2 A-F and Fig 3 A-H, please add.
- Please correct the order of the manuscript sections to the following: Abstract, Introduction, Results, Discussion, Methods, Acknowledgements, Disclosure and competing interests statement, References, Figure legends, Expanded View Figure legends.
- Rename "Materials and Methods" - it should be "Methods"
- Please note that the exact p values are not provided in the legends of figures 1C-E; 2I, 3D-G; 4C-F; 5B, C, D, G, H; 6D, F; 7E, F; 8I, J; 9D, E, J, K; 10C F; 11H, J; 12G, J K-M; 13E, F, J-M; EV1 B, C, F, G; EV2 C-F; EV3 3-H. Please provide exact values as reasonable.

I would like to suggest a few minor changes to the abstract that needs to be written in present tense. Please let me know whether you agree with the following:

Secretin is a gastrointestinal (GI) hormone that slows intestinal motility, an effect thought to be mediated through vagal afferent pathways. In this study we show evidence for a novel function of secretin involving a non-neural mechanism mediated by interstitial cells of Cajal (ICC). Transcripts of secretin receptors (Sctr) are expressed abundantly by ICC in the deep muscular plexus (ICC-DMP). Secretin inhibits small intestinal contractions in the presence of the neurotoxin, tetrodotoxin (TTX) and suppresses excitatory enteric neurotransmission. The inhibitory effects of secretin occur through inhibition of Ca²⁺ transients in ICC-DMP, likely via G α s-coupled cAMP production and PKA activation that leads to inhibition of IP₃ receptors. Our results provide a novel concept for the role of ICC-DMP in small intestinal motility. ICC-DMP serve as integration hubs in which signaling from the enteric nervous system and hormones converge and integrate regulatory responses controlling intestinal motility. In the case of secretin, integrated responses may serve to slow intestinal transit to enhance digestion and absorption of nutrients.

EMBO press papers are accompanied online by A) a short (1-2 sentences) summary of the findings and their significance, B) 2-3 bullet points highlighting key results and C) a synopsis image that is exactly 550 pixels wide and 200-600 pixels high (the height is variable). The synopsis image should provide a sketch of the major findings, like a graphical abstract. Please note that text needs to be readable at the final size. Please send us this information along with the final manuscript.

Best regards,

Esther

Referee #1:

Thank you for taking my comments into consideration and modifying the paper accordingly. I find the responses and the applied changes satisfying. I have noticed one small possible mistake:

Page 6, Line 1: I believe it is "Macaca" instead of murine

I do not ask for a minor revision as this (if I am right) can be corrected in the last step similarly to a spelling mistake.

Referee #2:

Thanks for addressing my remarks. I have no further comments.

Referee #3:

The authors have addressed my earlier comments, and I do not have any additional comments.

Dear Editor,

We thank the editor and reviewers for their support of our manuscript. We have addressed all editorial requests and Reviewer 1 request.

Sincerely,
Sal Baker

Referee #1:

Thank you for taking my comments into consideration and modifying the paper accordingly. I find the responses and the applied changes satisfying. I have noticed one small possible mistake:

Page 6, Line 1: I believe it is "Macaca" instead of murine

I do not ask for a minor revision as this (if I am right) can be corrected in the last step similarly to a spelling mistake.

Typo fixed.

Referee #2:

Thanks for addressing my remarks. I have no further comments.

Referee #3:

The authors have addressed my earlier comments, and I do not have any additional comments.

Prof. Salah Baker
University of Nevada, Reno
Physiology and Cell Biology
1664 N. Virginia ST
Reno, NV 89557
United States

Dear Prof. Baker,

I am very pleased to accept your manuscript for publication in the next available issue of EMBO reports. Thank you for your contribution to our journal.

Yours sincerely,
